# Decoding EEG With Spiking Neural Networks on Neuromorphic Hardware

**Neelesh Kumar**  *neelesh.kumar@rutgers.edu*
*Department of Computer Science*
*Rutgers University*

**Guangzhi Tang**  *guangzhi.tang@rutgers.edu*
*Department of Computer Science*
*Rutgers University*

**Raymond Yoo**  *raymond.yoo@rutgers.edu*
*Department of Computer Science*
*Rutgers University*

**Konstantinos P. Michmizos**  *michmizos@cs.rutgers.edu*
*Department of Computer Science*
*Rutgers University*

**Reviewed on OpenReview:** *https://openreview.net/forum?id=ZPBJPGX3Bz*

## Abstract

Decoding motor activity accurately and reliably from electroencephalography (EEG) signals is essential for several portable brain-computer interface (BCI) applications ranging from neural prosthetics to the control of industrial and mobile robots. Spiking neural networks (SNNs) is an emerging brain-inspired architecture that is well-suited for decoding EEG signals due to their built-in ability to integrate information at multiple timescales, leading to energy-efficient solutions for portable BCI. In practice, however, current SNN solutions suffer from i) an inefficient spike encoding of the EEG signals; ii) non-specialized network architectures that cannot capture EEG priors of spatiotemporal dependencies; and iii) the limited generalizability of the local learning rules commonly used to train the networks. These untapped challenges result in a performance gap between the current SNN approaches and the state-of-the-art deep neural network (DNN) methods. Moreover, the black-box nature of most current SNN solutions masks their correspondence with the underlying neurophysiology, further hindering their reliability for real-world applications. Here, we propose an SNN architecture with an input encoding and network design that exploits the priors of spatial and temporal dependencies in the EEG signal. To extract spatiotemporal features, the network comprised of spatial convolutional, temporal convolutional, and recurrent layers. The network weights and the neuron membrane parameters were trained jointly using gradient descent and our method was validated in classifying movement on two datasets: i) an in-house dataset comprising of complex components of movement, namely reaction time and directions, and ii) the publicly available eegmmidb dataset for motor imagery and movement. We deployed our SNN on Intel's Loihi neuromorphic processor, and show that our method consumed 95% less energy per inference than the state-of-the-art DNN methods on NVIDIA Jeston TX2, while achieving similar levels of classification performance. Finally, we interpreted the SNN using a network perturbation study to identify the spectral bands and brain activity that correlated with the SNN outputs. The results were in agreement with the current neurophysiological knowledge implicating the activation

patterns in the low-frequency oscillations over the motor cortex for hand movement and imagery tasks. Overall, our approach demonstrates the effectiveness of SNNs in accurately and reliably decoding EEG while availing the computational advantages offered by neuromorphic computing, and paves the way for employing neuromorphic methods in portable BCI systems.

# 1 Introduction

The accuracy and reliability in decoding electroencephalography (EEG) signals are the two main factors that brain-computer interface (BCI) relies on for its diverse set of applications, ranging from neurorehabilitation and neural prostheses (Baud et al., 2018; Chaudhary et al., 2016; Lebedev & Nicolelis, 2017), to the control of industrial (Buerkle et al., 2021; Garcia et al., 2013) and mobile robots (Bi et al., 2013). For their use to be practical in the real-world, BCI systems often need to be portable and run on limited energy resources (Zhang et al., 2019; Kartsch et al., 2019). Deep neural networks (DNNs) have been quite effective in decoding EEG for various tasks (LeCun et al., 2015) due to their ability to learn features from raw representations with strong generalization (Zhang et al., 2016). However, the high energy cost associated with inference using DNN (Dong et al., 2017) hinders their use in portable BCI.

Energy-efficiency is a long-promised advantage offered by spiking neural networks (SNN), an emerging brain-inspired architecture where neurons compute asynchronously through discrete events, the spikes. The hardware realization of the SNNs on neuromorphic chips (Davies et al., 2018) has spurred their applicability as energy-efficient solutions for many learning tasks (Tang et al., 2019; Hwu et al., 2017). Moreover, mounting evidence suggests that SNNs may be more effective than DNNs for the classification of signals that have non-stationary characteristics (Kugele et al., 2020; Deng et al., 2020; Reid et al., 2014). This has been attributed to the ability of the spiking neurons in representing information at multiple timescales through their dynamics (Deng et al., 2020; Yin et al., 2020; Bellec et al., 2018; Tang et al., 2021). However, current SNN-based EEG classification methods (Antelis et al., 2020; Luo et al., 2020; Kasabov, 2014) are unable to match the performance of the state-of-the-art DNN methods for EEG classification (Kumar & Michmizos, 2022; Schirrmeister et al., 2017), which renders SNNs ineffective for most practical applications.

The reported performance gap of SNNs is mainly due to three reasons. First, the common methods for encoding EEG into spikes (Schrauwen & Van Campenhout, 2003) are heuristics-based and require heavy task-independent tuning of encoding-related parameters. Consequently, an inappropriately tuned encoding of the spikes can significantly affect the classification performance. Second, several current approaches employ a reservoir of randomly connected recurrent neurons as their network topology (Antelis et al., 2020; Kasabov, 2014). While effective brain-inspired tuning methods for reservoirs have recently been introduced (Ivanov & Michmizos, 2021), they can only partially consider the important inductive priors associated with EEG in their design, such as the inherent spatial and temporal dependencies in the input signal (Burle et al., 2015). Last, local correlation-based learning rules (Markram et al., 1997) are very popular in training SNNs but they lack the expressiveness of error-driven training algorithms such as backpropagation, and restrict network topologies to shallow architectures (Shrestha et al., 2019). While learning rules such as spatiotemporal backpropagation (Wu et al., 2018) are now starting to gain traction in training SNNs for several tasks (Tang et al., 2021; 2020; Wu et al., 2018), they inherit the disadvantages found in their conventional counterparts, including interpretability. For example, the black-box nature of the SNN trained using backpropagation masks the correspondence between the learned features and the underlying neurophysiology, resulting in only weak guarantees for their reliability in analyzing EEG signals (Montavon et al., 2018). Therefore, the development of an accurate and reliable SNN solution to EEG classification requires a complete rethinking of the SNN design that can tackle the issues of input encoding, network topology, training, and interpretability.

In this work, we propose an SNN that extracts task-discriminative spatiotemporal EEG features by incorporating the priors of spatial and temporal dependencies in the brain activity directly in its input encoding and network design (Figure 1). The encoding parameters, network weights, and the neuron membrane parameters were jointly optimized using spatiotemporal backpropagation (Wu et al., 2018). We deployed the trained SNN on Intel's Loihi neuromorphic processor to demonstrate the energy efficiency for the inference. We validated our method on an in-house, IRB-approved, dataset for classifying complex movement compo-

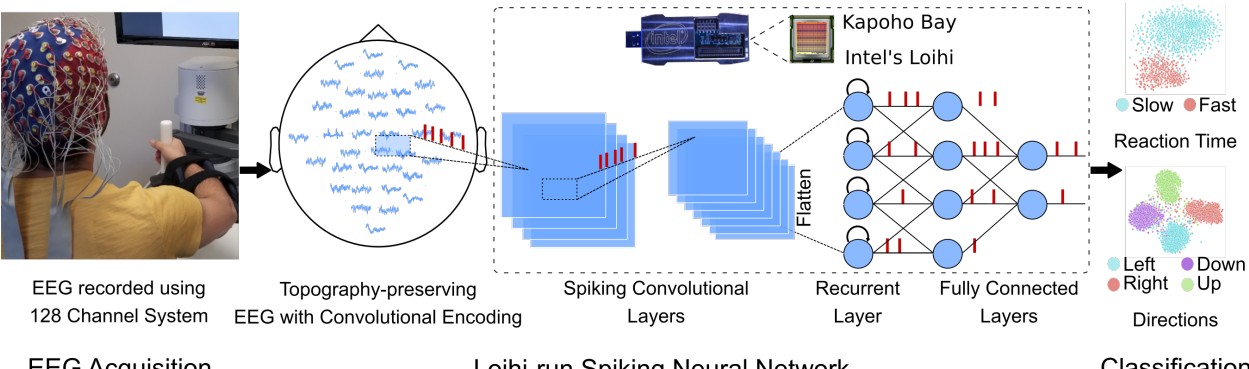

Figure 1: Experimental setup and study workflow: 12 subjects performed a goal-oriented motion task while high density EEG data were simultaneously acquired with movement kinematics. The proposed SNN received topography-preserving EEG inputs, and was deployed on Intel's Loihi neuromorphic chip to classify complex components of hand movements.

nents, namely reaction time (RT) and directions, and the publicly available eegmmidb dataset for classifying motor imagery and execution tasks. Our method on Loihi consumed 95% less energy per inference than the state-of-the-art DNN methods on NVIDIA Jetson TX2, while achieving similar levels of classification performance. To interpret the trained SNN, we used a network perturbation study and identified the low-frequency oscillations as well as the localized patterns of brain activity, in alignment with the current neurophysiological knowledge for motor tasks (Hamel-Thibault et al., 2018; Batula et al., 2017). Overall, our results demonstrate the effectiveness of SNN solutions in classifying temporal signals and discovering task-relevant features in the brain signals in an energy-efficient way. [1]

## 2 Related Works

Several studies have employed EEG to decode several behavior tasks. The earliest approaches relied on statistical methods that compute hand-crafted features such as spatial features (common spatial pattern (CSP) (Ang et al., 2008)), temporal features (fractal dynamics (Gupta et al., 2018)), or spectral features (wavelet transforms, power spectral density (Hazarika et al., 1997; Shaker, 2006; Kim et al., 2018)) from preprocessed EEG signals. A classifier such as a support vector machine (SVM) is then employed to segregate these features. While statistical EEG decoding methods are highly explainable, they require extensive preprocessing steps as well as domain-specific hand-crafted features, which results in high computational costs during inference and impedes their use in real-time systems. In addition, the reliance on a limited number of features affects their generalizability (Samek et al., 2012; Lotte & Guan, 2011) and as such, they do not classify well tasks that are important for decoding natural human behavior, such as hand movement directions (Robinson et al., 2012).

The recent DNN methods eliminate the need for domain-specific features by learning features directly from raw representations (LeCun et al., 2015) and also exhibit strong generalization (Zhang et al., 2016). The current DNN methods for EEG classification can be broadly categorized based on input representation and network design. Most existing approaches provide raw EEG representations with little or no preprocessing as inputs to the network. This type of input representation makes them suitable for network architectures such as 1D and 2D convolutional neural networks (1D-CNN (Acharya et al., 2018), 2D-CNNs (Schirrmeister et al., 2017; Cui et al., 2020; Lawhern et al., 2018; Gao et al., 2019)), or recurrent architectures such as LSTM (Zheng & Chen, 2021). While such approaches are truly end-to-end, the limited inductive priors in their input representation and network design limit their generalizability, and increase the amount of training data needed. Some recent approaches represent EEG inputs as 2D images either in the form of spectral

---

[1]Code available at https://github.com/combra-lab/snn-eeg

maps (Bashivan et al., 2015; Ieracitano et al., 2019; Jiao et al., 2018) or topological maps stacked in time to form 3D representations (Kumar & Michmizos, 2022; Zhao et al., 2019). Given the 3D input representations, these approaches employ convolutional-recurrent architectures such as CNN-LSTM (Bashivan et al., 2015) or 3D convolutional neural networks (3D-CNN) (Kumar & Michmizos, 2022; Zhao et al., 2019). Although DNN approaches like the ones based on 3D input representations are effective in classifying EEG signals for several tasks, their high energy consumption during inference (Dong et al., 2017) limits their use in applications that require portable BCIs.

SNNs provide highly energy-efficient solutions for many learning tasks (Davies et al., 2018; Tang et al., 2019). The current SNN-based approaches to EEG classification extract features from preprocessed EEG signals and feed it to an SNN-based multilayer perceptron (MLP) (Antelis et al., 2020) or CNN (Yan et al., 2022) for classification. These approaches suffer from the same problems found in traditional statistical methods, namely high computational costs during inference and limited generalizability. More recent approaches attempt to learn directly from the raw EEG signals by first encoding them into spikes using temporal or rate-based encoding (Schrauwen & Van Campenhout, 2003) and then feeding the spikes to a reservoir-style network architecture such as NeuCube for classification (Kasabov, 2014). Local learning rules (Markram et al., 1997) and brain-inspired tuning methods (Ivanov & Michmizos, 2021) are then be used to train these reservoirs in an unsupervised manner, and a biologically inspired supervised learning rule is used to train the readout layer for classification (Kasabov & Capecci, 2015; Tan et al., 2021; Alzhrani et al., 2021). These approaches have the appeal of being biologically relevant but they do not explore the full scale of learning rules present in the brain and therefore can perform well only in simple classification tasks.

With respect to interpretability, brain-inspired solutions such as NeuCube lend themselves well to neurophysiological interpretation, such as the identification of clusters of neurons that are implicated in a given cognitive task (Doborjeh et al., 2018; Kasabov & Capecci, 2015). Such interpretations are, however, lacking for hierarchical SNN architectures that are trained using backpropagation. While significant progress has been made in developing interpretation tools for DNNs (Selvaraju et al., 2017), such techniques cannot be directly employed for SNNs because they do not account for the temporal nature of SNN backpropagation (Kim & Panda, 2021). Overall, the use of heuristics-based encoding, local learning rules, and shallow network architectures limit the effectiveness and interpretability of the current approaches for complex EEG classification tasks.

## 3    Methods

We propose an SNN that incorporates the inductive priors of EEG spatial and temporal dependencies in its input encoding and network design. The encoding parameters, network weights, and neuron parameters were jointly trained using the spatiotemporal backpropagation learning algorithm (Wu et al., 2018). Then, we deployed the trained model on Loihi for energy-efficient inference. Finally, we interpreted the network using the network perturbation technique (Schirrmeister et al., 2017) to determine the EEG sensors and frequencies important for classification decisions.

### 3.1    Neuron Model

The building block of the SNN is the current-based leaky-integrate-and-fire (LIF) neurons (Lin et al., 2018). The states of the $k^{th}$ layer neurons at time $t$ were updated in two stages:

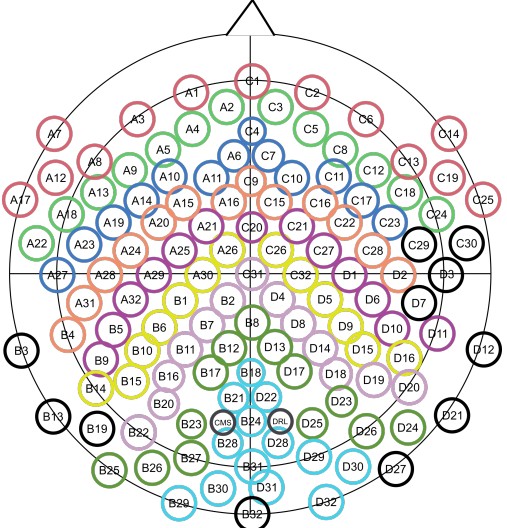

Figure 2: Input representations preserving the brain's topography applied on BioSemi's 128 sensor layout. Same colored sensors appeared in the same row in the input matrix. To ensure that the number of sensors per row are same, peripheral channels that were in close proximity to the picked channels were dropped. The dropped sensors are indicated in black.

First, the input spikes were integrated into synaptic current as

$$\mathbf{c}^{(t)(k)} = \mathbf{D}_c \cdot \mathbf{c}^{(t-1)(k)} + psp(\mathbf{o}^{(t)(k-1)}), \tag{1}$$

where $\mathbf{c}$ is the synaptic current, $\mathbf{D}_c$ is a non-negative vector of current decay factors for each neuron in the layer, $psp$ is the function that generates post-synaptic potential as defined by the $k^{th}$ layer, and $\mathbf{o}$ is an array of binary variable (0 or 1) which indicates the spike events of the presynaptic neurons.

Then, the synaptic current was integrated into membrane voltage and transformed to spikes as

$$\mathbf{v}^{(t)(k)} = \mathbf{D}_v \cdot \mathbf{v}^{(t-1)(k)} + \mathbf{c}^{(t)(k)}$$
$$o_i^{(t)(k)} = 1 \ \& \ v_i^{(t)(k)} = 0, \quad \text{if } v_i^{(t)(k)} > V_{th}^i(t), \tag{2}$$

where $\mathbf{v}$ is the membrane voltage, $\mathbf{D}_v$ is the non-negative vector of voltage decay factors for each neuron in the layer and $V_{th}$ is the voltage threshold which was either static or adaptive. To lessen the dependence on the initialization of hyperparameters and make them task dependent (Rathi & Roy, 2020), we set $\mathbf{D}_c$ and $\mathbf{D}_v$ as trainable parameters which were optimized using gradient descent. In addition, the adaptive firing thresholds were updated to increase after every spike and decay otherwise using

$$V_{th}^i(t) = V_{th}^i(t-1) + th_{amp}, \quad \text{if } o_i(t-1) = 1$$
$$V_{th}^i(t) = \max(V_{th}^i(t-1) \cdot th_{decay}, \ th_{base}), \quad \text{otherwise}, \tag{3}$$

where $th_{amp}$ is the amount by which voltage threshold increases , $th_{decay}$ is the decay factor for voltage threshold and $th_{base}$ is the minimum voltage threshold. Neurons with adaptive firing thresholds and decays are known to improve SNN performance for image classification tasks (Yin et al., 2020; Rathi & Roy, 2020) as they allow for multiscale temporal integration of information (Bellec et al., 2018). In our network, we set the thresholds for all the layers except the recurrent layer to be static. The recurrent layers used neurons with adaptive thresholds.

## 3.2 Spiking Neural Network for EEG Classification

The EEG data exhibit high spatial and temporal dependencies that are often correlated with the physical or mental activity being performed (Berka et al., 2007; Burle et al., 2015; Cohen & Kohn, 2011). Here, we propose an SNN that incorporated these dependencies in its architectural priors, enabling the extraction of task-discriminative spatiotemporal features for EEG classification.

The SNN exploited the spatial dependencies in the EEG data in three ways: through input representation, spike encoding, and network activity. We mapped the EEG signals from the spatially distributed channels onto a 2D matrix, leading to a topography-preserving input representation (Fig. 2). The mapped EEG signals were then fed to a convolutional spike encoding layer which converted them to spikes while preserving the spatial dependencies. As opposed to popular heuristics-based encoding algorithms for EEG (Petro et al., 2019), this direct approach made the encoding trainable and hence removed the need for manual tuning of encoding parameters. The encoded spikes were then passed through several layers of convolution in the SNN to extract spatial features (Equation 4). The post-synaptic potential function for the convolutional layer ($psp_H$) for the neuron at the location $(i, j)$ with the convolution channel $f$ in the $k^{th}$ layer is computed as

$$psp_{H_f}(\mathbf{o}^{(t)(k-1)})(i,j) = \sum_c \left( \sum_{m=0}^{M-1} \sum_{n=0}^{N-1} \mathbf{o}_c^{(t)(k-1)}(i-m, j-n) W_{H_{c,f}}^{(k)}(m,n) + b_{H_{c,f}}^{(k)} \right), \tag{4}$$

where $\mathbf{W}_{H_{c,f}}^{(k)}$ and $b_{H_{c,f}}^{(k)}$ are the kernel weights from convolution channel $c$ in layer $k-1$ to the channel $f$ in layer $k$. The convolution kernels are of the size $M$ x $N$. An average pooling layer between the convolutional layers downsampled the outputs for lessening the computational load and providing some translational invariance.

The SNN captured the multiscale temporal dependencies in the EEG data through the dynamics of the spiking neurons, a temporal convolution layer, and a recurrent layer. The dynamics of the spiking neurons enabled the network to learn weak short-term temporal dependence as they linearly integrated their past activities into their current activity (Equation 2). The temporal convolution layer and the recurrent layer in the SNN captured complex non-linear temporal relationships.

The temporal convolution layer performed non-linear integration of information from a short window of previous timesteps, allowing for learning of short-range temporal representations. Specifically, neurons at time $t$ integrate inputs from their presynaptic neurons over a window of previous timesteps from $t$ to $t - win$ into their post-synaptic potential ($psp_{TC}$) as defined by Equation 5.

$$psp_{TC}(\mathbf{o}^{(t)(k-1)}) = \sum_{dt=0}^{win} \mathbf{W}_{TC}^{(k)(dt)}\mathbf{o}^{(t-dt)(k-1)} + \mathbf{b}_{TC}^{(k)(dt)},$$

(5)

where $\mathbf{W}_{TC}^{(k)(dt)}$, $\mathbf{b}_{TC}^{(k)(dt)}$ are the weights and biases for each time $dt$ in the window.

The recurrent layer captured complex non-linear temporal relationships over larger timescales. Specifically, spikes of a neuron from the previous timestep were fed to it as an input in the current timestep which were then integrated into their post-synaptic potential ($psp_R$) as per Equation 6.

$$psp_R(\mathbf{o}^{(t)(k-1)}) = \mathbf{W}^{(k)}\mathbf{o}^{(t)(k-1)} + \mathbf{b}^{(k)} + \mathbf{W}_R^{(k)}\mathbf{o}^{(t-1)(k)} + \mathbf{b}_R^{(k)},$$

(6)

where $\mathbf{W}^{(k)}$, $\mathbf{b}^{(k)}$ are the feedforward and $\mathbf{W}_R^{(k)}$, $\mathbf{b}_R^{(k)}$ are the recurrent weights and biases.

Finally, the spatiotemporal features extracted by the SNN were fed to a decoder module composed of a spiking and a non-spiking fully-connected layer which transformed them into a vector of class probabilities at each timestep according to Equation 7.

$$psp_O(\mathbf{o}^{(t)(K-1)}) = \mathbf{W}^{(K)}\mathbf{o}^{(t)(K-1)} + \mathbf{b}^{(K)}$$
$$\mathbf{p}^{(t)} = \mathbf{W}^{out}\mathbf{o}^{(t)(K)} + \mathbf{b}^{out},$$

(7)

where $\mathbf{W}^{(K)}$ and $\mathbf{b}^{(K)}$ denote the weights and biases of the spiking fully-connected layer $K$ receiving inputs from layer $K-1$, and $\mathbf{W}^{out}$ and $\mathbf{b}^{out}$ are the weights and biases of the non-spiking fully-connected layer. A weighted sum of the probabilities at each timestep $\mathbf{p}^{(t)}$ yielded the final class probabilities $\mathbf{p}_{class}$ (Equation 8)

$$\mathbf{p}_{class} = \sum_t W_{ts}^{(t)} \cdot \mathbf{p}^{(t)},$$

(8)

where $W_{ts}^{(t)}$ is the decoding weight for each timestep. The decoding weights on the timesteps allowed the network to learn the relative importance of each timestep towards classification (Appendix C).

Figure 3: **A.** The proposed SNN architecture. Topography-preserving inputs were fed to convolutional layers for extracting spatial features. Temporal features were extracted using temporal convolutional and recurrent layers. **B.** Transformation of neuron states by a spiking neuron unit.

### 3.3 Network Training

The SNN parameters and the neuron decays ($\mathbf{D}_c$, $\mathbf{D}_v$) were trained to minimize the average cross-entropy loss $L$ over the training samples. The gradients of the parameters with respect to $L$ were computed using the spatiotemporal backpropagation algorithm (Wu et al., 2018). To handle the non-differentiability of the

threshold function, we used the rectangular function (Wu et al., 2018) to approximate the gradient of a spike with respect to the membrane voltage $v$

$$z(v) = \begin{cases} a_1 & \text{if } |v - V_{th}| < a_2 \\ 0 & \text{otherwise} \end{cases} \tag{9}$$

where $z$ is the pseudo-gradient, $a_1$ is the amplifier, $a_2$ is the threshold window for passing the gradient.

The gradient of the loss with respect to the neuron states- current, voltage, and output spikes is given by:

$$\nabla_{\mathbf{v}^{(t)(k)}} L = z(\mathbf{v}^{(t)(k)}) \cdot \nabla_{\mathbf{o}^{(t)(k)}} L + d_v(1 - \mathbf{o}^{(t)(k)}) \cdot \nabla_{\mathbf{v}^{(t+1)(k)}} L$$
$$\nabla_{\mathbf{c}^{(t)(k)}} L = \nabla_{\mathbf{v}^{(t+1)(k)}} L + d_c \nabla_{\mathbf{c}^{(t+1)(k)}} L \tag{10}$$
$$\nabla_{\mathbf{o}^{(t)(k-1)}} L = \mathbf{W}^{(k)'} \cdot \nabla_{\mathbf{c}^{(t)(k)}} L.$$

Using this, we compute the gradient of the loss with respect to the network parameters. Specifically, the gradients for the spatial convolution and linear layers are

$$\nabla_{\mathbf{W}^{(k)}} L = \sum_{t=1}^{T} \mathbf{o}^{(t)(k-1)} \cdot \nabla_{\mathbf{c}^{(t)(k)}} L \quad , \quad \nabla_{\mathbf{b}^{(k)}} L = \sum_{t=1}^{T} \nabla_{\mathbf{c}^{(t)(k)}} L, \tag{11}$$

and the gradients of the temporal convolution layer are

$$\nabla_{\mathbf{W}_{TC}^{(k)(dt)}} L = \sum_{t=win+1}^{T} \mathbf{o}^{(t-dt)(k-1)} \cdot \nabla_{\mathbf{c}^{(t)(k)}} L \quad , \quad \nabla_{\mathbf{b}_{TC}^{(k)(dt)}} L = \sum_{t=win+1}^{T} \nabla_{\mathbf{c}^{(t)(k)}} L. \tag{12}$$

The gradients for the recurrent layer are

$$\nabla_{\mathbf{W}_{R}^{(k)}} L = \sum_{t=2}^{T} \mathbf{o}^{(t-1)(k)} \cdot \nabla_{\mathbf{c}^{(t)(k)}} L \quad , \quad \nabla_{\mathbf{b}_{R}^{(k)}} L = \sum_{t=2}^{T} \nabla_{\mathbf{c}^{(t)(k)}} L. \tag{13}$$

Finally the gradients for the decays are

$$\nabla_{\mathbf{D}_c} L = \sum_{t=1}^{T} \mathbf{c}^{(t-1)(k)} \cdot \nabla_{\mathbf{c}^{(t)(k)}} L \quad , \quad \nabla_{\mathbf{D}_v} L = \sum_{t=1}^{T} \mathbf{v}^{(t-1)(k)} \cdot (1 - \mathbf{o}^{(t-1)(k)}) \cdot \nabla_{\mathbf{v}^{(t)(k)}} L. \tag{14}$$

### 3.4 Neuromorphic Realization

We deployed our trained SNN on Intel's Loihi neuromorphic processor (Davies et al., 2018) for energy-efficient inference. To maximize the inference speed, we introduced an interaction framework between Loihi and the host machine that minimized the communication overhead during each inference. The convolutional spike encoding layer was implemented on the host machine, with the generated spikes sent to Loihi in spike-time buffers to avoid redundant communications at every timestep. The SNN decoding module was implemented on Loihi. To do so, we utilized the low-frequency x86 cores that is embedded on Loihi for interfacing with the on-chip networks during runtime. The sequence of operations were as follows: First, the x86 chip computed the weighted sum of output spikes $\mathbf{o}^{(t)(K)}$ as per Equation 15,

$$\mathbf{o}_{sum}^{(K)} = \sum_{t} W_{ts}^{(t)} \cdot \mathbf{o}^{(t)(K)} \tag{15}$$

Then, the computed sum was sent to the host machine to be converted to final class probabilities through the non-spiking fully-connected decoding layer as follows,

$$\mathbf{p}_{class} = \mathbf{W}^{out} \mathbf{o}_{sum}^{(K)} + \mathbf{b}^{out}, \tag{16}$$

Table 1: List of hyperparameters

| Hyperparameters | Reaction Time | Directions | EEGMMIDB | EEGMMIDB w/Loihi |
|---|---|---|---|---|
| Segmentation intervals | -0.5-0s | -0.5-1.5s | 0-1s | 0-1s |
| Initial voltage decay ($\mathbf{D}_v$) | 0.1 | 0.1 | 0.1 | 0.1 (Fixed) |
| Initial current decay ($\mathbf{D}_c$) | 0.1 | 0.1 | 0.1 | 0.1 (Fixed) |
| Initial voltage threshold ($V_{th}$) | 0.1 | 0.1 | 0.1 | 0.2 (Fixed) |
| Feedforward timesteps | 125 | 500 | 160 | 160 |
| Convolutional layer architecture | 64C1-128C2-256C3 | 64C1-128C2-256C3 | 64C1-128C2-256C3 | 4C1-8C2-128C3 |
| Timesteps for temporal convolutional($win$) | 3 | 3 | 3 | 3 |
| Pseudograd amplifier ($a_1$) | 0.3 | 0.3 | 0.3 | 0.3 |
| Pseudograd window ($a_2$) | 0.3 | 0.3 | 0.3 | 0.3 |
| Weight learning rate | 1e-4 | 1e-4 | 1e-4 | 1e-4 |
| Decays learning rate | 1e-3 | 1e-3 | 1e-4 | 1e-4 |
| Batch size | 64 | 64 | 64 | 64 |

where the $\mathbf{p}_{class}$ computed here is mathematically equivalent to the definition in Equation 8.

We used layer-wise rescaling to map the full-precision weights of the trained SNN to the 8 bits integer weights on Loihi (Tang et al., 2020). Each convolution operation defined in Equation 4 had a separate group of physical connections on Loihi, with all groups sharing the same weights. The temporal convolution operation was realized through spike delays which Loihi supports in the form of buffered presynaptic spikes on the neuron cores. Due to hardware limitations, the SNN realized on Loihi did not have neurons with adaptive thresholds and decays.

### 3.5  Network Perturbation

To interpret the features that the SNN used in its decision-making, we performed a network perturbation study for correlating changes (perturbations) in the spectral bands of the EEG inputs with the network outputs (Schirrmeister et al., 2017). Doing so allowed us to determine the causal effect of changing the band-power on the network output. To perturb the spectral bands of the EEG trials, we first converted the trials into the frequency domain using Fourier transformation. Then, we added a small Gaussian noise (with mean 0 and variance 1) to the amplitudes while keeping the phases unchanged. Next, we transformed the signals back into the time domain using inverse Fourier transform, which gave us the perturbed EEG trials. Then, we computed the network output for each layer for the unperturbed and perturbed trials. Finally, we measured the correlation in the change in the network outputs with the change in the spectral amplitudes.

## 4  Experiments and Results

To demonstrate the effectiveness of our method, we applied it to motor decoding tasks and validated it on two datasets: i) an in-house dataset consisting of complex components of arm movements performed on a plane, and ii) the public eegmmidb dataset (Schalk et al., 2004) for motor imagery and movement. Then, we performed a network perturbation study to demonstrate the neurophysiological interpretability of our method. Lastly, we deployed the trained networks on Loihi to highlight the energy efficiency of our method.

### 4.1  Data Preprocessing

We preprocessed the EEG data to get rid of the contaminations. The preprocessing was minimal to allow the SNN to learn the representations by itself. To remove the low and high-frequency artifacts and drifts, we applied a zero-phase FIR bandpass filter of 0.1Hz-40Hz (for the in-house dataset) and 0.1Hz-80Hz (for eegmmidb). We then applied independent component analysis (ICA) (Radüntz et al., 2015) to the data from each subject to get rid of ocular artifacts. To do so, we applied the FastICA (Hyvärinen & Oja, 2000) algorithm to construct 25 ICA components. The components corresponding to ocular artifacts were then manually identified and removed. Subsequently, we segmented the data into trials containing the events of interest. The time windows used for segmentation for the different classification tasks are provided in Table 1. For the in-house dataset, we applied baseline correction from the beginning of the data until the time

Table 2: Two-class classification of Reaction Time evaluated using leave-one-subject-out (%).

| Subject | 3D-CNN | 2D-CNN | CNN-TC | CNN-LSTM | EEGNet | SNN (Ours) |
|---------|--------|--------|--------|----------|--------|------------|
| 1 | 80.00 | 67.30 | 82.81 | 67.85 | 72.22 | 87.03 |
| 2 | 79.68 | 65.85 | 77.78 | 70.58 | 78.57 | 80.35 |
| 3 | 81.03 | 69.64 | 84.37 | 54.68 | 71.15 | 84.61 |
| 4 | 82.25 | 81.96 | 76.47 | 70.58 | 73.17 | 75.61 |
| 5 | 79.68 | 73.21 | 73.21 | 69.41 | 78.68 | 80.32 |
| 6 | 79.51 | 58.53 | 75.00 | 70.58 | 73.43 | 81.53 |
| 7 | 74.51 | 77.58 | 79.31 | 65.95 | 74.11 | 75.29 |
| 8 | 80.00 | 79.31 | 82.75 | 67.21 | 79.68 | 92.30 |
| 9 | 73.91 | 76.78 | 84.37 | 64.06 | 79.31 | 82.75 |
| 10 | 85.33 | 72.41 | 84.12 | 74.60 | 82.81 | 89.06 |
| 11 | 81.57 | 82.35 | 80.95 | 60.93 | 70.21 | 76.59 |
| 12 | 80.26 | 80.32 | 80.85 | 73.43 | 80.95 | 77.81 |
| Average | $79.81 \pm 3.07$ | $73.77 \pm 7.34$ | $80.17 \pm 3.82$ | $67.49 \pm 5.55$ | $76.19 \pm 4.24$ | $81.93 \pm 5.45$ |

point zero to get rid of unrelated temporal drifts. Lastly, we normalized the trials after segmentation using z-score normalization with statistics computed across all the trials of training data. More details on the preprocessing steps are provided in Appendix B.

## 4.2 Datasets, Baselines and Evaluation

We performed an IRB-approved experiment in which 12 subjects performed goal-directed arm movements on a plane in response to visual stimuli using BioNik's InMotion Arm Rehabilitation Robot. The experiment protocol is described in Appendix A. High density EEG data were simultaneously acquired using a 128-sensors Biosemi ActiveOne EEG system at 1024Hz., which was later downsampled to 250Hz. The resulting dataset consisted of i) Reaction time (RT), measured as the time difference between the onset of a stimulus and the start of the movement. We discretized the RT into two classes- fast and slow, by choosing suitable thresholds determined from the histogram of RTs for each subject separately, based on the distribution of their RT across the experiment. The RT dataset consisted of 50 concurrent EEG trials on average per class for each subject; ii) Directions: We separated the data into 4 classes representing the 4 orthogonal directions - left, right, up, and down movements. This dataset consisted of 52 concurrent EEG trials, per direction, for each subject.

The publicly available eegmmidb dataset comprised of EEG recordings from 109 subjects acquired using a 64-channel system during motor movement and imagery tasks (Schalk et al., 2004). Subjects were required to move or imagine moving their fists or feet in response to visual cues. We trained the SNN to classify whether the movement was performed using the left hand, or the right hand, or the feet, for both real and imagined movements. Each class had 2524 EEG trials on average.

We compared our method against the following state-of-the-art DNN methods:

1. A 3D-CNN comprising of 3D convolutional layers to extract spatiotemporal features from EEG inputs with topographical representation (Kumar & Michmizos, 2022).

2. A 2D-CNN comprising of several layers of 2D convolutions to extract spatiotemporal features from stacked EEG inputs (Kumar & Michmizos).

3. A CNN with separate temporal and spatial convolutions (CNN-TC) (Schirrmeister et al., 2017)

4. A CNN-LSTM where the CNN learns the spatial features and the LSTM learns the temporal features (Zhang et al., 2018).

5. A compact convolutional network, EEGNet, consisting of temporal convolution, depthwise convolution, and separable convolution followed by pointwise convolution (Lawhern et al., 2018).

Table 3: Four-class classification of Directions evaluated using leave-one-subject-out (%)

| Subject | 3D-CNN | 2D-CNN | CNN-TC | CNN-LSTM | EEGNet | SNN (Ours) |
|---------|--------|--------|--------|----------|--------|------------|
| 1 | 82.85 | 76.19 | 82.85 | 53.33 | 80.00 | 83.81 |
| 2 | 83.81 | 74.28 | 76.19 | 70.47 | 80.95 | 87.71 |
| 3 | 72.38 | 63.81 | 69.52 | 56.19 | 72.38 | 69.52 |
| 4 | 84.76 | 80.00 | 82.85 | 54.28 | 81.90 | 82.85 |
| 5 | 77.14 | 67.61 | 74.28 | 46.66 | 68.57 | 80.95 |
| 6 | 72.38 | 59.04 | 65.71 | 49.52 | 64.76 | 73.33 |
| 7 | 81.20 | 73.15 | 74.49 | 51.67 | 77.18 | 83.22 |
| 8 | 87.61 | 71.42 | 82.85 | 70.47 | 81.90 | 83.81 |
| 9 | 83.81 | 61.90 | 64.76 | 42.85 | 70.47 | 68.57 |
| 10 | 85.71 | 59.04 | 75.28 | 30.47 | 74.28 | 83.81 |
| 11 | 95.23 | 86.66 | 82.87 | 42.85 | 92.38 | 86.67 |
| 12 | 77.14 | 67.61 | 73.33 | 57.14 | 69.52 | 75.23 |
| Average | $82.00 \pm 6.24$ | $70.05 \pm 8.53$ | $75.41 \pm 6.54$ | $52.16 \pm 10.79$ | $76.19 \pm 7.68$ | $79.95 \pm 6.28$ |

Given that the baselines employed different EEG systems requiring network architecture changes, we implemented our own versions of the baselines. The hyperparameters for the baselines and the proposed method were optimized using a held-out set consisting of data from 1 subject for the in-house dataset, and 9 subjects for the eegmmidb dataset. We used the leave-$k$-out technique for evaluation, where data from all but $k$ subjects were used for training. The evaluation was done on the data from the $k$ left-out subjects. This evaluation technique allowed us to evaluate the ability of the SNN to generalize to new subjects that were not included in the training. To ensure that all the subjects were covered for evaluation, we trained different networks with different folds of $k$ subjects left out from training. We then computed the average of the classification accuracies obtained from all the folds. For the in-house dataset, $k = 1$, i.e. there were 11 training subjects and 1 test subject. For the eegmmidb dataset, $k = 10$, which meant that there were 90 training subjects and 10 test subjects. The list of hyperparameters for training is provided in Table 1.

### 4.3 Classification of Complex Movement Components

When evaluated using the leave-$k$-out technique ($k = 1$), our SNN achieved comparable performance as the DNN methods on both the RT and directions classification tasks, achieving average accuracy of $81.93\% \pm 5.45\%$ for RT (Table 2) and $79.95\% \pm 6.28\%$ for directions (Table 3). To further highlight the importance of the different components in the SNN architecture, we performed an ablation study where we a) removed the recurrent and temporal convolutional layers in the network ("No recurrent & TC") while keeping all the other components intact ; b) removed the recurrent layer in the network ("No recurrent") while keeping all the other components intact; and c) trained the network without adaptation of decays or voltage thresholds ("No adaptation") while keeping all the other components intact. When compared against our SNN (Table 4), the networks with "No recurrent & TC", "No recurrent", and "No adaptation" suffered from a drop in accuracies, thereby highlighting the importance of these components towards classification.

### 4.4 Classification of limb for Motor Execution and Imagery

For further validation, we evaluated the SNN on the publicly available eegmmidb dataset which comprises of EEG recordings from 109 subjects acquired using a 64-channel system during motor movement and imagery tasks (Schalk et al., 2004). Subjects were required to move or imagine moving their fists or feet in response to visual cues. We trained the SNN to classify whether the movement was performed using the left hand, or the right hand, or the feet. When evaluated using the leave-$k$-out technique ($k = 10$), the SNN achieved similar levels of performance as the state-of-the-art DNN methods (Table 5). Overall, our method successfully bridged the performance gap between existing SNN methods and the DNN methods, thereby suggesting SNN as a possible alternative to DNN for EEG classification. Additional classification results are presented in Appendix E.

Table 4: Ablation: Classification of Directions (%)

| Subject | No recurrent & TC | No recurrent | No adaptation | SNN (Ours) |
|---|---|---|---|---|
| 1 | 75.23 | 78.09 | 81.90 | 83.81 |
| 2 | 78.09 | 80.95 | 84.76 | 87.71 |
| 3 | 54.28 | 65.70 | 63.81 | 69.52 |
| 4 | 78.09 | 80.00 | 80.95 | 82.85 |
| 5 | 60.09 | 64.76 | 79.04 | 80.95 |
| 6 | 59.04 | 67.61 | 69.52 | 73.33 |
| 7 | 68.45 | 77.85 | 80.53 | 83.22 |
| 8 | 74.28 | 79.19 | 81.90 | 83.81 |
| 9 | 55.23 | 65.71 | 72.38 | 68.57 |
| 10 | 60.95 | 75.23 | 76.19 | 83.81 |
| 11 | 70.47 | 85.71 | 85.71 | 86.67 |
| 12 | 72.38 | 74.28 | 76.19 | 75.23 |
| Average | $67.21 \pm 8.82$ | $74.59 \pm 7.01$ | $77.74 \pm 6.48$ | $79.95 \pm 6.28$ |

Table 5: Classification of Motor Movement/Imagery evaluated using leave-10-subjects-out (%)

| Task | Real or Imagined | 3D-CNN | 2D-CNN | CNN-TC | CNN-LSTM | EEGNet | SNN (Ours) |
|---|---|---|---|---|---|---|---|
| Hand (L) vs. Hand (R) | Real | $82.68 \pm 3.22$ | $78.51 \pm 3.45$ | $78.37 \pm 3.92$ | $75.48 \pm 3.11$ | $74.64 \pm 4.84$ | $80.27 \pm 3.87$ |
| Hand (L) vs. Feet | Real | $77.95 \pm 3.99$ | $75.53 \pm 4.67$ | $74.80 \pm 4.81$ | $70.72 \pm 4.65$ | $70.17 \pm 2.94$ | $77.69 \pm 4.00$ |
| Hand (R) vs. Feet | Real | $77.76 \pm 4.11$ | $74.63 \pm 2.43$ | $72.55 \pm 4.83$ | $71.21 \pm 3.89$ | $70.23 \pm 1.78$ | $74.74 \pm 3.51$ |
| Hand (L) vs. Hand (R) | Imagined | $81.02 \pm 2.72$ | $79.84 \pm 3.75$ | $78.86 \pm 3.63$ | $74.42 \pm 3.15$ | $76.02 \pm 3.28$ | $80.65 \pm 3.83$ |
| Hand (L) vs. Feet | Imagined | $75.60 \pm 3.55$ | $74.62 \pm 3.94$ | $73.77 \pm 2.69$ | $70.95 \pm 4.29$ | $70.81 \pm 2.19$ | $74.33 \pm 3.90$ |
| Hand (R) vs. Feet | Imagined | $73.48 \pm 1.66$ | $73.07 \pm 1.89$ | $71.70 \pm 2.17$ | $69.67 \pm 4.13$ | $70.55 \pm 1.94$ | $72.31 \pm 1.79$ |

## 4.5 Network Interpretation using a Perturbation Study

Neuroimaging studies have shown that motor actions are associated with spectral power modulations at multiple frequency levels (Jasper & Penfield, 1949; Jeon et al., 2011; Pfurtscheller et al., 2006). Hence, EEG band power features are useful in the decoding tasks. To determine if the SNN used these features in its decision-making, we performed a network perturbation study to correlate the changes in the spectral bands with the network outputs (see Methods). The network perturbation study was performed on the eegmmidb dataset since more domain knowledge is available for the corresponding classification tasks, allowing us to validate our findings.

We averaged the correlation between spectral perturbation and network output (see Methods) for left and right motor cortex electrodes, and each frequency range (Delta: 1-4 Hz, Theta: 5-8 Hz, Alpha: 9-13 Hz, Beta: 14-30 Hz) over all the trials of a representative validation subject, resulting in a spectral map (Fig. 4 Left). The averaged correlation values were then rescaled to be in the range $[-1, 1]$ for facilitating comparisons across different frequency bands and across different subject groups. The raw absolute maximum correlation values are presented in Appendix D. For obtaining the topographical maps, we repeated the above procedure for all the electrodes. The computed topographical maps (Fig. 4 Right) showed localized regions that the network possibly exploited in its decision-making.

All spectral bands showed some degree of correlation for the movement and imagery tasks, with the delta band being the most prominent (Fig. 4). This is in alignment with recent studies implicating the role of low-frequency oscillations in predicting handedness of the movement (Hamel-Thibault et al., 2018), and in movement preparation (Caspers et al., 1984; Birbaumer et al., 1990) and control (Tarkka & Hallett, 1990; Yilmaz et al., 2015). In addition, the topography of the delta band correlations suggest localized patterns in both the movement and imagery tasks. Specifically, for the movement task, an amplitude decrease in the right sensorimotor area led to an increase in the prediction of the Hand (L) class and vice-versa for the Hand (R) class. This localized pattern was previously observed experimentally in Hamel-Thibault et al. (2018); Hoshi & Tanji (2006); Beurze et al. (2009) for hand selection in reaching tasks, and also for DNNs

in Schirrmeister et al. (2017). The opposite effect was observed in the case of imagery tasks, suggesting differences in the activation patterns of the two tasks, consistent with the observations in (Batula et al., 2017). These patterns were less distinct for the alpha band, potentially due to the weaker correlation of the network's output with perturbation in the alpha band. Nevertheless, we see hint of similar patterns in the lower motor cortex for alpha band. However, as noted in Schirrmeister et al. (2017), the perturbation results reflect the behavior of the trained model, and any interpretation about the data itself must be carefully made.

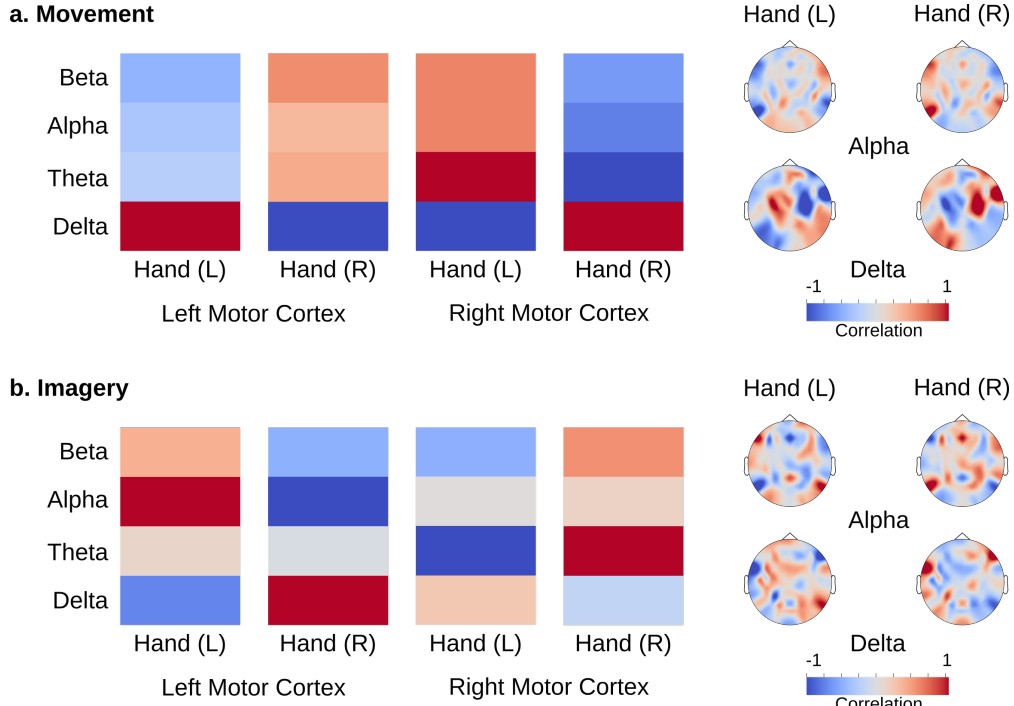

Figure 4: Neurophysiological network interpretation. The spectral maps obtained using the network perturbation study identified all frequencies as being correlated with the movement tasks, with delta being the most prominent (**a.** Left, **b.** Left). The topographical maps of the correlations reveal contralateral activation pattern for the movement task (**a.** Right), and opposite pattern for the imagery task (**b.** Right). The maps are averaged over the trials from all the test subjects in one fold. The maps for the remaining folds are provided in Appendix D.Correlation values are scaled to be in the range $[-1, 1]$.

### 4.6   Energy-efficient Inference on Loihi

For energy-efficient inference, we deployed the SNN trained for Hand (L) vs. Hand (R) (Real) task on Loihi. Due to hardware limitations and to allow for high inference speed, the SNN on Loihi had reduced number of convolutional kernels (Table 1) and did not have neurons with adaptive threshold and decays. This resulted in a marginal decrease in the performance of the SNN on Loihi when compared against SNN on a full precision machine (Table 6). We computed the inference speed and energy consumption for EEG inference of the SNN on Loihi, and compared them against the SNN and DNN methods on the embedded AI chip NVIDIA Jetson TX2 (operating in energy-efficient mode Q). We computed the energy cost per inference as the ratio of the dynamic power and the number of inferences per second. Our SNN on Loihi consumed 95% less energy per inference than the 3D-CNN on the low-power processor for DNNs- Jetson TX2 (Table 6). It is worth noting that the encoding and decoding computations that took place outside of the Loihi constitute only 2.9% of the overall computations, and hence do not contribute to the energy costs much.

Table 6: Performance across network architectures and hardware for Hand (L) vs. Hand (R) (Real)

| Network (Device) | SNN (Loihi) | SNN (TX2) | 2D-CNN (TX2) | 3D-CNN (TX2) |
|---|---|---|---|---|
| Accuracy (%) | $78.04 \pm 3.25$ | $79.40 \pm 4.03$ | $78.51 \pm 3.45$ | $82.68 \pm 3.22$ |
| Static Power (W) | 1.639 | 1.606 | 1.672 | 1.691 |
| Dynamic Power (W) | 0.0367 | 4.871 | 4.040 | 4.481 |
| Speed (Inference/s) | 10.2 | 1.38 | 73.2 | 59.5 |
| Energy (mJ/Inference) | 3.60 | 3529.71 | 55.20 | 75.31 |

## 5 Discussion and Conclusion

In this work, we demonstrated that through the synergy of a) an appropriate input encoding, b) network topology that reflected data priors, and c) a powerful learning algorithm, SNNs can reach the high performance of the DNNs in modeling intrinsically variable EEG signals, while being orders of magnitude more energy efficient than their mainstream counterparts. To that effect, our SNN extracted task-discriminative spatiotemporal EEG features for motor imagery and movement classification and achieved similar levels of performance as the state-of-the-art DNN methods. The energy-efficient inference on Loihi further demonstrated the practical advantages of the emerging neuromorphic technology. Moreover, the correspondence of the learned features with the underlying neurophysiology increased the reliability of our method, allowing it to be placed in real-world portable BCI systems.

This work tackles an important gap in SNN approaches, that has hindered them from becoming a mainstream technology and replace or complement DNNs. While SNNs are arguably better suited than DNNs for spatiotemporal tasks (Kugele et al., 2020; Deng et al., 2020; Reid et al., 2014), there has remained a significant performance gap between SNN and DNN for EEG classification (Antelis et al., 2020; Luo et al., 2020; Kasabov, 2014). We showed here that an appropriately constructed SNN that incorporates the high spatiotemporal dependencies in its input encoding and network design through specialized layers (convolutional, temporal convolutional, and recurrent), and uses gradient-based learning rules as opposed to local learning rules could achieve similar levels of performance as the DNN methods. This is evident through the high leave-one-out accuracy achieved by our method, thus highlighting its practical effectiveness in decoding inherently noisy EEG data.

Further practical gains of our method is demonstrated through neuromorphic realization, where our method consumed 95% less energy per inference than the DNN methods when deployed on Loihi while obtaining comparable performance levels. The significance of this result lies in applications such as EEG-based portable BCI systems, which run on limited onboard power sources (Zhang et al., 2019; Kartsch et al., 2019). The use of DNN methods running on conventional microprocessors can quickly deplete the energy resources on such devices. On the other hand, our proposed method can arguably decrease the high energy costs by a large factor for such applications. In fact, further improvements in energy efficiency can be achieved by employing memristive neuromorphic processors (Ankit et al., 2017).

The decisions taken by the SNNs must be supported by the domain knowledge for the method to be reliable enough for use in real-world applications. It is common knowledge that an unreliable SNN may use incorrect features to make its decisions, and achieve high accuracy on the limited validation set while exhibiting unintended consequences when deployed in the real-world (Sturm et al., 2016). That is why the interpretability of our SNN was central in this work. We interpreted the SNN using a perturbation study that identified the low-frequency oscillations and localized patterns of activations, both of which align well with the current neurophysiological knowledge for motor tasks (Hamel-Thibault et al., 2018; Batula et al., 2017). The correspondence of the learned features with the underlying neurophysiology increases the reliability of our method to be placed in critical real-time systems, including the ones for enhancing functional motor recovery (Chaudhary et al., 2016; Donoghue, 2018).

This paper describes our efforts in reducing the performance gap between SNNs and DNNs for EEG classification. To this end, we suggested novel network architectures and learning rules that reflect the inductive biases in EEG. Further performance improvements may be achieved by exploiting the advances in the training

of DNN. Some of these include the development of SNN-equivalent of normalization layers such as Batch-Norm for stabilizing the training (Ioffe & Szegedy, 2015); attention-modules for capturing arbitrary lengths of spatial and temporal dependencies (Vaswani et al., 2017); and learning rules that more closely approximate ANN backpropagation gradients. Efforts in these directions are already being made with architectures that incorporate normalization layers (Zheng et al., 2020), temporal attention modules (Yao et al., 2021), and learning rules that do not rely on surrogate gradients (Wu et al., 2021). The enhanced performance, in combination with the interpretability, can make our method directly deployable to any real-world task that involves decoding brain behavior from the inherently noisy EEG signals. Overall, our results demonstrate the effectiveness of SNN solutions in classifying EEG signals by discovering task-relevant features in the recorded brain activity, that can be harvested in an energy-efficient way.

## Acknowledgements

This work is supported through the Grant K12HD093427 from the National Center for Medical Rehabilitation Research, NIH/NICHD; and Intel's Neuromorphic Research Community Grant Award.

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
