# OpenReview forum: "Decoding EEG With Spiking Neural Networks on Neuromorphic Hardware"
_TMLR — Accepted by TMLR_

### Review · Reviewer_1m1u · 2022-05-11

**Summary Of Contributions:**

The paper proposes a novel spiking neural network architecture and training pipeline for decoding of motor movement related EEG data. Experiments are performed on both the EEGMMIDB dataset for motor imagery/execution classification, as well as an in-house dataset for classification of hand movement direction and reaction times. Authors deployed their discriminative SNN pipeline on Intel’s Loihi neuromorphic processor to realistically demonstrate energy-efficient inference. Finally a Fourier domain perturbation based network interpretability method is explored with demonstrations in accordance with neurophysiological evidence.

**Requested Changes:**

1) Details with regards to where static or adaptive thresholds were used is a bit unclear in the manuscript. Were they only static for the Loihi deployment phase? Another question for instance: following topography-preserving EEG mapping, a convolutional spike encoding layer is utilized. What was the firing threshold following this layer such that a sufficient amount of incoming spikes could be provided to the encoder block? Was it simply fixed to a constant value, and the learnable encoding layer alleviated any related issues?

2) Could you please elaborate how essential was the use of a non-spiking fully-connected W_{out} layer at the end of the network? Did the authors explore alternative output decoding schemes, such as sum of spikes from o^{(t)(K)}?

3) Can the authors provide more details to the ICA-based ocular artifact removal procedure, as it is currently superficially phrased in the manuscript? Did it involve manual IC selection? How many ICs were constructed initially by the algorithm? Did the authors use pooled ICA across subjects, or ICA applied on each subject’s data per experiment?

4) Please elaborate the phrase: “we normalized the segmented trials using z-score normalization”. Does it actually mean that z-score normalization was applied to trials after segmenting?

5) For the reaction time and direction experiments, was there any baseline correction applied to EEG segments?

6) The paper lacks details on the in-house dataset specifications, and these can be included in depth. How was the experimental paradigm set? With regards to some relevant work, e.g., did it involve center-out reaching movements within a circle [1,2]?
[1] “Classification of different reaching movements from the same limb using EEG”, Journal of Neural Engineering, 2017.
[2] “Electroencephalographic identifiers of motor adaptation learning”, Journal of Neural Engineering, 2017.

7) Since data analyses involve cross-subject EEG classification, it naturally brings up the question: did the authors use any kind of EEG transfer learning or stronger model regularization to impose representation invariance in EEG decoding for the baseline networks? For instance the CNN-TC architecture explicitly benefits from stronger dropout in this setting [Schirrmeister et al., 2017].

8) Identifying relevance of low-frequency EEG components for reaction times would align well with previous similar evidence on movement-related cortical potentials (MRCPs). However Figure 4 analyses are performed on the EEGMMIDB motor imagery/execution data of a single subject. In this case one would perhaps expect results to show relevances in the alpha or beta bands for motor imagery/execution [Pfurtscheller et al.]. For a sanity check, it would be clarifying to also present how the other frequency band topographies look like (i.e., similar to the right side of Figure 4 where only delta-band results are shown)? Were there similar findings for some other subjects as well? Would model interpretations look similar for different models (e.g., CNN-TC) than the proposed SNN?

Minor comments:
- Typo in Table 1: “Psuedograd” -> “Pseudograd”.
- Please add in captions of Tables 2 and 3 the information on these results being “k=1 fold” or “leave-one-subject-out”, and being averaged over 12 repetitions, and the number of classes per classification task.

**Strengths And Weaknesses:**

Strengths: I sincerely thank the authors for their nice submission. The paper has a clear perspective, structure and narrative. Neuromorphic realization experiments is an important strength of this work. Overall it was a nice read.

Weaknesses: There are some arising questions with regards to the experimental pipeline, interpretability results, and EEG data analyses that can be clarified in revisions. My comments/questions for the authors’ revisions are listed below.

---

> ### Author Response · Authors · 2022-05-31
> **Response to reviewer 1m1u (1/2)**
>
> **Q.1.** Details with regards to where static or adaptive thresholds were used is a bit unclear in the manuscript. Were they only static for the Loihi deployment phase?
>
> **A.1.**  We clarify that the use of adaptive thresholds was only for recurrent layer neurons for the non-Loihi phase. For all other layers, our network had neurons with static thresholds. Moreover, for the Loihi phase, all the layers, including the recurrent layer, had neurons with static thresholds. We now better clarify the use of adaptive thresholds in our revised manuscript (Section 3.1; Third para).
>
> **Q.2.** Another question for instance: following topography-preserving EEG mapping, a convolutional spike encoding layer is utilized. What was the firing threshold following this layer such that a sufficient amount of incoming spikes could be provided to the encoder block? Was it simply fixed to a constant value, and the learnable encoding layer alleviated any related issues?
>
> A.2. The reviewer is indeed right in assuming that the firing threshold for the encoding block was set to a constant value, and the learnable encoding layer ensured that a sufficient amount of spikes was generated.
>
> **Q.3.** Could you please elaborate how essential was the use of a non-spiking fully-connected W_{out} layer at the end of the network? Did the authors explore alternative output decoding schemes, such as sum of spikes from o^{(t)(K)}?
>
> **A.3.** We thank the reviewer for their well-thought-out question. The use of non-spiking fully connected layer with weights on the timesteps was essential because it helped network training in two ways: i) the non-spiking module provides more information in the gradients for training than simply the sum of spikes; ii) the weights on the timesteps allow the network to learn the importance of each timestep towards classification, as opposed to sum of weights which assume that all timesteps hold equal importance. We also explored alternative output decoding schemes, in particular the sum of spikes mentioned by the reviewer. Specifically, we trained an SNN for the classification of left hand vs. right hand (Real) using the sum of spikes decoding scheme. The sum of spike decoding scheme obtained 73.57\% accuracy, as opposed to 80.27\% obtained by our method. Per the reviewer’s suggestion, we elaborate on the importance of our decoding scheme in Section 3.2 of the revised manuscript.
>
> **Q.4.** Can the authors provide more details to the ICA-based ocular artifact removal procedure, as it is currently superficially phrased in the manuscript? Did it involve manual IC selection? How many ICs were constructed initially by the algorithm? Did the authors use pooled ICA across subjects, or ICA applied on each subject’s data per experiment?
>
> **A.4.** We agree with the reviewer that the description of ICA was inadequate. We revised the manuscript to add more details about our ICA-based ocular artifact removal (Section 4.1). Briefly, we applied ICA to data from each subject using the FastICA[1] method. The number of components was set to 25, and the components corresponding to ocular artifacts were identified and removed manually. We note that the manual ICA method can be automated using techniques such as [2] to enable real-time usage of our method.
>
> [1] Hyvärinen, Aapo, and Erkki Oja. "Independent component analysis: algorithms and applications." Neural networks 13.4-5 (2000): 411-430.
>
> [2] Chang, W.-D., Lim, J.-H. & Im, C.-H. An unsupervised eye blink artifact detection method for real-time electroencephalogram processing. Physiol. Meas. 37, 401 (2016).
>
> **Q.5.** Please elaborate the phrase: “we normalized the segmented trials using z-score normalization”. Does it actually mean that z-score normalization was applied to trials after segmenting?
>
> **A.5.** We clarify that the z score normalization was indeed applied to trials after segmentation. The normalization was done across all the trials based on the statistics computed from the training data. We now clarified its description in the revised manuscript (Section 4.1).
>
> **Q.6.** For the reaction time and direction experiments, was there any baseline correction applied to EEG segments?
>
> **A.6.** We clarify that baseline correction was indeed applied from the beginning of the data until time zero. We now include this detail in the Data Preprocessing section (Section 4.1).

---

> > ### Author Response · Authors · 2022-05-31
> > **Response to reviewer 1m1u (2/2)**
> >
> > **Q.7.** The paper lacks details on the in-house dataset specifications, and these can be included in depth. How was the experimental paradigm set? With regards to some relevant work, e.g., did it involve center-out reaching movements within a circle [1,2]? [1] “Classification of different reaching movements from the same limb using EEG”, Journal of Neural Engineering, 2017. [2] “Electroencephalographic identifiers of motor adaptation learning”, Journal of Neural Engineering, 2017.
> >
> > **A.7.** We thank the reviewer for their suggestion of including details about the experimental paradigm. We now describe the experimental protocol in Appendix A. The task included center-out reaching movements similar to the references brought up by the reviewer. Briefly, we developed a visually-guided goal-directed motion task and asked the subjects to perform it on the arm rehabilitation robot. The task environment is comprised of a pointer and a target box. The pointer indicated the current position of the end-effector of the robotic arm in the 2D plane of motion. After the pointer entered the target box, the next target box appeared with a delay of 20ms with an added jitter sampled from a uniform distribution in the range [-10ms, 10ms]. The target appeared at random in any of the four orthogonal directions- left, right, up, or down.
> >
> > **Q.8.** Since data analyses involve cross-subject EEG classification, it naturally brings up the question: did the authors use any kind of EEG transfer learning or stronger model regularization to impose representation invariance in EEG decoding for the baseline networks? For instance the CNN-TC architecture explicitly benefits from stronger dropout in this setting [Schirrmeister et al., 2017].
> >
> > **A.8.** We clarify that for the baseline model CNN-TC, we indeed used dropout. In fact, the model architecture for all the baselines was the same as described in the respective papers, with the training hyperparameters tuned for the specific tasks.
> >
> > **Q.9.** Identifying relevance of low-frequency EEG components for reaction times would align well with previous similar evidence on movement-related cortical potentials (MRCPs). However Figure 4 analyses are performed on the EEGMMIDB motor imagery/execution data of a single subject. In this case one would perhaps expect results to show relevances in the alpha or beta bands for motor imagery/execution [Pfurtscheller et al.]. For a sanity check, it would be clarifying to also present how the other frequency band topographies look like (i.e., similar to the right side of Figure 4 where only delta-band results are shown)? Were there similar findings for some other subjects as well? Would model interpretations look similar for different models (e.g., CNN-TC) than the proposed SNN?
> >
> > **A.9.** We thank the reviewer for their suggestion on enhancing our perturbation results. As per the reviewer’s suggestion, we have now modified Figure 4 to include topographical maps from the alpha band frequencies as well, in addition to delta band frequency. As hinted by the reviewer, the maps for the alpha band look similar to that of the delta band. Additionally, to indicate group results, we now show averaged maps over trials from all the test subjects in a fold (e.g. subjects 1-10 in fold 1). The averaged maps for subjects in all other folds are provided in Appendix B. With this approach, we are able to show the results for all the subjects which are in fact consistent with each other. In addition, the model interpretation does indeed look similar for CNN-TC (Figure 8 in [1]).
> >
> > [1] Schirrmeister, Robin Tibor, et al. "Deep learning with convolutional neural networks for EEG decoding and visualization." Human brain mapping 38.11 (2017): 5391-5420.

---

> > > ### Comment · Reviewer_1m1u · 2022-06-09
> > > **Thanks for your replies**
> > >
> > > A.1. & A.2. Alright, this clarifies my question. Thanks for the edits.
> > >
> > > A.3. Thanks for the clarification, then this appears to be very important for learning. It should be explicit in the manuscript. Perhaps you can put a "(see Suppl.)" type of statement to your text addition in Sec 3.2, and present a short table in Suppl. with these sum-of-spike type output coding ablations with the lower 73.57%.
> > >
> > > A.4. Thanks for the clarification.
> > >
> > > A.5. Okay, thanks for rephrasing.
> > >
> > > A.6. Thanks for the addition. I just assumed this one, but this is new information which was not in the manuscript previously. Please keep in mind that one has to describe the data processing pipeline step by step in detail (in the Suppl.), and clearly, for reproducibility.
> > >
> > > A.7. Thanks for the clarification. Please include a figure illustration of this paradigm in the Appendix too, potentially with references (later) in which this dataset was maybe used.
> > >
> > > A.8. Thanks. This appears to be also in line with the new EEGNet results as far as I see.
> > >
> > > A.9. Thanks for the additions on this. To be clear, I do not see any direct similarity between the alpha and delta band topographies in Figure 4 of the manuscript. Same holds for your response on the similarity with CNN-TC (Fig 8 in [1]), which is not clear. That being said, subject grouped resulting figures in the Appendix B seem pretty self-consistent to me for the delta band results, which is nice.

---

> > > > ### Author Response · Authors · 2022-06-09
> > > > **Updated response to Reviewer 1m1u**
> > > >
> > > > **[On Decoding module]**: We thank the reviewer for their suggestion. We now present the comparison with sum-of-spike decoding in Table 1 in the supplementary material with a pointer to it in the main text in Sec 3.2.
> > > >
> > > > **[On Data preprocessing pipeline]**: We thank the reviewer for their suggestion, and have added more details on the full pipeline of our data preprocessing in the supplementary material.
> > > >
> > > > **[On Experimental paradigm]**: Per the reviewer’s suggestion, we have now added a figure illustration of the paradigm in the supplementary material. We will add references to the works where the dataset has been previously used when the submission can be deanonymized.
> > > >
> > > > **[On Network Interpretation]**: We thank the reviewer for their thoughtful remark. We agree that the patterns of correlations for the alpha band are not as distinct as the delta band. We think that this is because the network’s correlation with the delta band is stronger than that with the alpha band (Fig 4). That said, one may observe some hint of contralateral patterns in the lower motor cortex similar to delta band patterns. We now mention these caveats in the revised manuscript (Section 4.5; Third para).
> > > >
> > > > For comparison with Figure 8 in the [Schirrmeister et al. 2017], we agree with the reviewer that the band correlations are different from ours. This may be because the dataset in [Schirrmeister et al. 2017] was collected in a setting that optimized the acquisition of high frequency components, because of which these components exhibit high correlation in Fig. 8. However, we can still see that the patterns of correlations in the motor cortex presented in the right panel in Fig 8 are similar to our Fig 4 in our paper. This is in fact the only observation that we claim in our paper (Section 4.5; third para).

---

> > > > > ### Comment · Reviewer_WM3g · 2022-06-10
> > > > > **correlation scale for colorbar**
> > > > >
> > > > > Especially after averaging it seems surprising that the correlation values approach ±1. I am guessing this visualization is not showing the correlation coefficients, but some scaled version perhaps by dividing by the maximum absolute value of the observed correlation coefficients. In Schirrmeister et al.'s paper (2017) the maximum is closer to 0.8.
> > > > >
> > > > > Does the colorbar range accurately reflect the correlation coefficient?

---

> > > > > > ### Author Response · Authors · 2022-06-10
> > > > > > **Clarifying correlation scale for colorbar**
> > > > > >
> > > > > > The reviewer is right in assuming that the correlation values have been normalized to be in the range [-1,1]. This was done to facilitate easier comparison across different frequency bands and groups of subjects. We realize that it can cause confusion because the real range is not reflected in the colorbar, and we apologize for it. We now present the actual max absolute correlation values in Table 2 in Appendix D for each group of subjects (max correlation being 0.67 for test fold 5). We also clarify this in the revised manuscript (Section 4.5; Para 2).

---

### Review · Reviewer_WM3g · 2022-05-12

**Summary Of Contributions:**

The paper describes an approach for using spiking neural network for EEG decoding that will be much more efficient during inference. In particular, the results show that the predication performance (accuracy for discrete decoding) is comparable to current convolutional networks for EEG decoding but can run on specialized hardware for EEG at much higher efficiency. The paper describes an approach that is meant to exploit both the spatiotemporal patterns in EEG.  In covers some aspects of spiking neural network model and training. The contribution is mainly the architecture choices specific to EEG decoding.

**Broader Impact Concerns:**

Nothing is particularly concerning assuming the IRB for human subjects is valid.  More broadly, efficient EEG decoding may enable some technology that is potentially beneficial for creating brain-computer interfaces. However, their could be deleterious impacts to such technology too.

**Requested Changes:**

Clarifications (answers are critical to recommend acceptance):
In Section 3.1, it is not clear if $\mathbf{D}_c$ and $\mathbf{D}_v$ have constraints (non-negativity perhaps). If so then how are these constraints enforced during the gradient descent.

In Equation 4  it is not clear how the neuron at location (i,j) mentioned before the equation relates to $A(i,j)$ in the equation, as after the equation $A$ is described as a convolutional kernel. Where is the input in equation 4? Is the input treated as the previous layer?

The $psp(\cdot)$ appears on the left hand side in three different equations,  and it is not clear how to reconcile equations 4, 5, and 6. Are they perhaps supposed to use layer indexes of k from disjoint sets?

Equation 7 may have some indexing errors as there doesn't seem to be a statement relating the parenthetical superscript K-1 in the first line with the K on the second line.

In Section 3.3., is there any particularly special about the gradients as given. Besides the first line of equation 10 that involves the $z$ function, these seem to be standard and it is not clear if there inclusion is necessary.  What is different from spatiotemporal back propagation of Wu et al. 2018?

In Section 3.4, I find the mix of hardware details a bit obscure. What is meant by "low-frequency x86 chip". Is that one of the processors on the Loihi system on chip? A full hardware description and how this affects the reported energy usage and computation is necessary to understand the import of this statement.

In section 4.1, details of the bandpass filter (order and type) should be given. Also how were the ICA components associated to ocular artifacts identified, manually? For time windows of multivariate data, the use of z-score normalization is ambiguous. Is this done per trial, across trails (based on only training data), at each time point in the window? It is also difficult because it is not clear how the time points in the window are used by the SNN or DNN.

The results in 4.3 and 4.4 are those shown in 4.2. The organization and presentation of the results is confusing. Basic information about the datasets (number of channels, number of training subjects, number of trials, validation set choice and usage, etc. should be reported in 4.2 before Tables 2 and 3. Additionally the class distribution and/or chance or naive prediction rate.

In 4.2, the baseline methods' hyper parameters are said to have used "a separate held-out set". It is not clear if that is the same as the proposed method and also how that is used in conjunction with the leave-k-out technique.  In Section 4.4, k=10 for the leave k out. In this case how is hyper-parameter selection via validation performed?

As the interpretability was done for a single subject, it is unclear how consistent this is. How was the subject selected? Can the interpretation maps for all validation subjects be shown? From a neurophysiological perspective it would be reasonable that they should be roughly the same.  (Also validation trials is mentioned in the Figure 4 caption, which is confusing given the hold-k subjects out method).

In the last paragraph of Section 5, please clarify what aspects of the learning rules were novel and important in the context of EEG.

Minor:
In Equation 3 please typeset max in roman font and use another symbol instead of asterisk for scalar multiplication to preclude confusion with convolution.

In Section 4.3 please place quotes or use different typesetting for "No current and temporal conv" and "No adaptation" in text.

Mathematical equations and expressions should be treated as part of the normal syntax. Commas after equations if they are followed by dependent clauses and periods (full stops) at the conclusion of sentences. There is often no need to use a colon before them in the case "We define something as:".

Bibliography style: Please capitalize acronyms consistent with normal usage. Journal names, conference venues, and book title should be capitalized according to title case.

**Strengths And Weaknesses:**

Strengths:
The rational for exploiting spatiotemporal patterns for EEG in a spiking neural network is reasonable. The related work seems applicable.  The efforts towards interpretation seems to be readily comprehensible. The energy efficiency results are very promising.

Weaknesses:
The presentation of the network is not clear in how the input is fed to the network.  Everything is given as a function of the previous layer and many equations seem to overload the definition of the post-synaptic potential (psp).

The specific novel contributions are not clear. Not that novelty is not sufficient but to clarify what was previous work from other authors, especially for the spatiotemporal backpropogation learning rule.

The hyper-parameter selection in the context of leave-k-out for subjects is not discussed. Basic statistics of sample sizes etc. should be included.

---

> ### Author Response · Authors · 2022-05-31
> **Response to reviewer WM3g (1/3)**
>
> **Q.1.** Dc and Dv have constraints (non-negativity perhaps). If so then how are these constraints enforced during the gradient descent.
>
> **A.1.** We thank the reviewer for their thoughtful question. Although the decays are supposed to be positive, we clarify that we did not have any constraints on Dc and Dv during their training. Rather, we initially set them to positive values and set the learning rates to low values (1e-3 and 1e-4) because of which the values after training remain positive.
>
> **Q.2.** In Equation 4 it is not clear how the neuron at location (i,j) mentioned before the equation relates to  A(i,j) in the equation, as after the equation A is described as a convolutional kernel. Where is the input in equation 4? Is the input treated as the previous layer? The psp(⋅) appears on the left hand side in three different equations, and it is not clear how to reconcile equations 4, 5, and 6. Are they perhaps supposed to use layer indexes of k from disjoint sets?
>
> **A.2.** We agree with the reviewer that our notations to describe the layer output were confusing. We have revised the psp notations in equations 4, 5, 6 in our manuscript to indicate the layer to which they belong. We clarify that the inputs to the layers are just the outputs from the previous layer. However, to keep the equations general and not tied down to a single architecture, we do not assume anything about the layer from which these inputs are coming. We further clarify that equation 4 describes the psp for the kth layer neuron located at (i, j) with the convolution channel f as a function of the input spikes and the convolution kernel A. We have revised the text in the manuscript (Section 3.2; Para 2) to more clearly describe equation 4.
>
> **Q.3.** Equation 7 may have some indexing errors as there doesn't seem to be a statement relating the parenthetical superscript K-1 in the first line with the K on the second line.
>
> **A.3.** We clarify that K here refers to the number of layers in the network. Accordingly, K-1 is the layer right before the decoding layer. We have clarified this in our revised manuscript (Section 3.2; Equation 7).
>
> **Q.4.** In Section 3.3., is there any particularly special about the gradients as given. Besides the first line of equation 10 that involves the z function, these seem to be standard and it is not clear if there inclusion is necessary. What is different from spatiotemporal back propagation of Wu et al. 2018?
>
> **A.4.** We clarify that the difference from [Wu et al. 2018] is that we describe the flow of gradients for the specialized layers (convolutional, temporal convolutional, and recurrent), and for the trainable decays, which is missing in [Wu et al. 2018]. In addition, our network consisted of a current-based leaky integrate and fire neuron model, because of which gradients were computed not just with respect to voltage, but also current.
>
> **Q.5.** In Section 3.4, I find the mix of hardware details a bit obscure. What is meant by "low-frequency x86 chip". Is that one of the processors on the Loihi system on chip? A full hardware description and how this affects the reported energy usage and computation is necessary to understand the import of this statement.
>
> **A.5.** We thank the reviewer for their suggestion of including hardware details. We now provide the hardware details of Loihi in our revised manuscript (Section 3.4). Briefly, the low-frequency x86 cores are the processors on the periphery of Loihi that allow it to interface with the on-chip networks during runtime. We used the x86 cores to implement the decoding module which reduced the data transfer load between Loihi and the host machine. In terms of energy usage, the measurements we report are for the computations on the Loihi chip and the x86 processor. The computations that take place outside of the chip (encoding and decoding) constitute only 2.9% of the overall computations (as measured using the number of addition and multiplication operations), and hence do not contribute to the energy costs much. We clarify this in section 4.6 of the revised manuscript.

---

> > ### Author Response · Authors · 2022-05-31
> > **Response to reviewer WM3g (2/3)**
> >
> > **Q.6.** In section 4.1, details of the bandpass filter (order and type) should be given.
> >
> > **A.6.** We thank the reviewer for their suggestion, and now provide the details of the bandpass filter in section 4.1 in the revised manuscript. To clarify, we used a zero-phase FIR filter with the filter length chosen automatically based on the size of the transition regions (https://mne.tools/stable/generated/mne.filter.filter_data.html).
> >
> > **Q.7.** Also how were the ICA components associated to ocular artifacts identified, manually?
> >
> > **A.7.** We agree with the reviewer that our description of ICA was inadequate. We have revised the manuscript to add more details about our ICA-based ocular artifact removal (Section 4.1). Briefly, we applied ICA to data from each subject using the FastICA[1] method. The number of components was set to 25, and the components corresponding to ocular artifacts were identified and removed manually. We also note that the manual ICA method can be automated using techniques such as [2] to enable real-time usage of our method.
> >
> > [1] Hyvärinen, Aapo, and Erkki Oja. "Independent component analysis: algorithms and applications." Neural networks 13.4-5 (2000): 411-430.
> >
> > [2] Chang, W.-D., Lim, J.-H. & Im, C.-H. An unsupervised eye blink artifact detection method for real-time electroencephalogram processing. Physiol. Meas. 37, 401 (2016).
> >
> > **Q.8.** For time windows of multivariate data, the use of z-score normalization is ambiguous. Is this done per trial, across trails (based on only training data), at each time point in the window? It is also difficult because it is not clear how the time points in the window are used by the SNN or DNN.
> >
> > **A.8.** We agree with the reviewer that our description of the z-score normalization was unclear. We now clarify that the z-score normalization was done across all the trials based on the statistics computed from the training data at each time point. That is, the mean and standard deviation used for normalization formed a vector across all the time steps in the window. We further clarify that the EEG data at each time point in the window formed input to the SNN at each timestep. For the DNN, the data from the entire window was fed to the network in one shot since there is no temporal component in the DNN.
> >
> > **Q.9.** The results in 4.3 and 4.4 are those shown in 4.2. The organization and presentation of the results is confusing. Basic information about the datasets (number of channels, number of training subjects, number of trials, validation set choice and usage, etc. should be reported in 4.2 before Tables 2 and 3. Additionally the class distribution and/or chance or naive prediction rate.
> >
> > **A.9.** We thank the reviewer for their suggestion on improving the organization of our experiments and results. As per the reviewer’s suggestion, we now report the information about both the datasets- number of channels, training and validation subjects, and number of trials per class (to indicate class distribution) in section 4.3 in the revised manuscript.
> >
> > **Q.10.** In 4.2, the baseline methods' hyperparameters are said to have used "a separate held-out set". It is not clear if that is the same as the proposed method and also how that is used in conjunction with the leave-k-out technique. In Section 4.4, k=10 for the leave k out. In this case how is hyper-parameter selection via validation performed?
> >
> > **A.10.** We apologize for the unclear description of our training, test, and validation strategy. We believe that the confusion arose due to our interchangeable use of “test” and “validation” trials. We divided the dataset into train and test using the leave-k-out strategy, meaning that data from all but k subjects formed the training dataset, and the data from the left out k subjects formed the test dataset. To ensure that all the subjects were covered for evaluation, we trained different networks with different folds of $k$ subjects left out from training. In addition, we used a separate group of subjects (validation subjects) to tune the hyperparameters of both the baselines and our proposed method. We have now revised the description of our data partitioning strategy in section 4.2 for better clarity.

---

> > > ### Author Response · Authors · 2022-05-31
> > > **Response to reviewer WM3g (3/3)**
> > >
> > > **Q.11.** As the interpretability was done for a single subject, it is unclear how consistent this is. How was the subject selected? Can the interpretation maps for all validation subjects be shown? From a neurophysiological perspective it would be reasonable that they should be roughly the same. (Also validation trials is mentioned in the Figure 4 caption, which is confusing given the hold-k subjects out method).
> > >
> > > **A.11.** We thank the reviewer for their well-thought-out comment. As per the reviewer’s suggestion, we have now modified Figure 4 to show averaged maps over trials from all the test subjects in a fold (subjects 1-10 in fold 1). The averaged maps for subjects in all other folds are provided in Appendix B. With this approach, we are able to show the results for all the subjects. As suggested by the reviewer, they are indeed roughly the same for all folds.
> > >
> > > **Q.12.** In the last paragraph of Section 5, please clarify what aspects of the learning rules were novel and important in the context of EEG.
> > >
> > > **A.12.** We clarify that in the context of EEG, we are the first work to propose the use of a gradient-based learning rule for training an end-to-end SNN. Prior works have focussed on local learning rules, which although biologically plausible, lack the expressiveness of backpropagation. As per the reviewer’s suggestion, we clarify the distinction with the prior works in section 5 (Para 2) of the revised manuscript.
> > >
> > > **Q.13.** In Equation 3 please typeset max in roman font and use another symbol instead of asterisk for scalar multiplication to preclude confusion with convolution. In Section 4.3 please place quotes or use different typesetting for "No current and temporal conv" and "No adaptation" in text. Mathematical equations and expressions should be treated as part of the normal syntax. Commas after equations if they are followed by dependent clauses and periods (full stops) at the conclusion of sentences. There is often no need to use a colon before them in the case "We define something as:". Bibliography style: Please capitalize acronyms consistent with normal usage. Journal names, conference venues, and book title should be capitalized according to title case.
> > >
> > > **A.13.** We thank the reviewer for their careful scrutiny of our text and have made the recommended changes in our revised manuscript.

---

> > > > ### Comment · Reviewer_WM3g · 2022-06-05
> > > > **check standard typesetting**
> > > >
> > > > I think the edits actually made the typesetting of mathematical equations worse. Please read https://www.jmlr.org/format/format.html.
> > > >
> > > > The comma or period/full stop should be at the end of the expression, not at the new line following. There is no need to use colon in most cases.

---

> > > > > ### Author Response · Authors · 2022-06-06
> > > > > **Updated Response to Reviewer WM3g**
> > > > >
> > > > > **[Constraints on decays]** As per the reviewer’s suggestion, we now clamp the decays in the code after every optimization update so that they stay positive. This does not change our results since our decays never go negative. We state the non-negative constraints on these values in the revised manuscript (Section 3.1).
> > > > >
> > > > > **[Notation in equation 4]** We thank the reviewer for their careful scrutiny of our equations, and for their suggestion to make it more clear. We have rewritten equation 4 in the revised manuscript for better clarity, and answer specific questions below:
> > > > > - **On covolutional kernel and difference between W and A**: In our revised equation, we condensed A and W into a single matrix W which indicates the kernel weights from the convolution channel c in layer k-1 to the channel f in layer k. We also unwrapped the convolution operation to make the role of the convolution kernel more clear.
> > > > > - **On the range of indices**: We clarify that the indices i, j indicate the neurons at the coordinates of the spatial location in the 2D matrix input representation, and their range is same as the dimensions of that matrix. These are analogous to (but not the same as) the indices i in equation 2 which describe the variables for a general vector of neurons, and aren’t architecture specific (which is why we do not specify the range).
> > > > >
> > > > > **[Held-out set]** We thank the reviewer for pointing out that the description of held-out set is lacking. We answer their questions below:
> > > > >
> > > > > - **Provenance and sample size**: The separate held-out validation set consisted of data from 1 subject for the in-house dataset (208 samples for Directions; 105 samples for RT), and 9 subjects for the eegmmidb dataset (200 samples on average for each classification task). These subjects were different from the ones used in training and evaluation. We have now added these details in the revised manuscript (Section 4.2; Third para).
> > > > > - **How was it used**: In the tuning phase, data from all subjects except the held-out set were used for training the networks. We monitored the accuracy on the held-out set to tune the hyperparameteres of the proposed method and the baselines.
> > > > > - **Was the average leave-one or leave-k out used?**: Since the held-out set was only one fold of subjects, average leave-one or leave-k were not used.
> > > > >
> > > > > **[Typesetting of mathematical equations]** We thank the reviewer for their careful scrutiny, and have incorporated their suggestion on typesetting mathematical equations in our revised manuscript.

---

> > > ### Comment · Reviewer_WM3g · 2022-06-06
> > > **"the" separate held-out set is not clear**
> > >
> > > What is the provenance, sample size, and target task for the separate held-out set? How was it used? Was the average leave-one or leave-k out used?

---

> > ### Comment · Reviewer_WM3g · 2022-06-05
> > **notation in equation 4**
> >
> > This notation still doesn't make sense to me. Is A(i,j)  a scalar or a matrix? Is there a different convolutional kernel at each location? What is different between the weights of a convolution layer (denoted W with various scripts) and A? Perhaps this equation is typeset out of order, but I don't understand its syntax.
> >
> > Is the range of $i$ in equation 2, which should be given, the same range as the $i$ and $j$ in equation 4?

---

> > ### Comment · Reviewer_WM3g · 2022-06-05
> > **constraints on values supposed to be positive**
> >
> > I would suggest that rather than set the learning rate to a low value and hope the values remain positive, that these constraints be stated and incorporated in the code...

---

### Review · Reviewer_Gshi · 2022-05-12

**Summary Of Contributions:**

The manuscript describes a novel spiking neural network architecture for EEG decoding. The SNN architecture is evaluated in terms of accuracy and energy usage on EEG movement-decoding tasks. The manuscript claims 95% reduction in energy usage for the SNN implemented on Loihi and comparable decoding accuracy compared to regular convolutional neural networks.

**Broader Impact Concerns:**

No Concerns.

**Requested Changes:**

1. Comparisons to published results
2. Add EEGNet to comparisons
3. Provide code
4. Clarify energy efficiency comparison
5. Ablation only removing recurrence
6. Discuss visualization caveats
7. Add projection details


**Strengths And Weaknesses:**

The manuscript describes an interesting SNN architecture. The main issues to me are that the evaluation on public datasets/comparison to prior work is limited and I could not see a statement about code availability?

1. In my view, to ensure that the research community can properly assess the decoding performance of spiking neural networks, there need to be direct comparisons to published results. The manuscript however only provides comparisons to reimplemented baselines “Given that the baselines employed different EEG systems requiring network architecture changes, we implemented our own versions of the baselines and optimized the hyperparameters using a separate held-out set.”... I would much favor to still see comparison to published results, on eegmmidb or also BCI Competition IV  datasets, e.g. BCI Competition IV 2a… This would greatly help assess if/how big of a performance gap there is, since the compared results will have been well-tuned.

2. Why are the comparisons on eegmmidb only binary and not multiclass?

3. Code availability is always a plus as it allows other researchers to apply the proposed model to other datasets.

4. Regarding the energy-efficiency/energy comparisons, I am not so sure how to interpret them due to: “ The convolutional spike encoding layer was implemented on the host machine, with the generated spikes sent to Loihi in spike-time buffers to avoid redundant communications at every timestep. The SNN decoding module was implemented on Loihi.” Is this common practice that some part of the model is not taken into account when computing energy consumption? Isn’t this comparison unfair then, assuming e.g. 2d-CNN all the energy is taken into account, while for SNN some part is left out? How big is this part, the convolutional spike encoding layer? Some clarifications here would be helpful.

5. Also “SNN on Loihi had reduced number of convolutional kernels (Table 1) and did not have neurons with adaptive threshold and decays.”.. Shouldn’t this then also be done for the SNN outside of Loihi? Also for “We used layer-wise rescaling to map the full-precision weights of the trained SNN to the 8 bits integer weights on Loihi (Tang et al., 2020).”

6. Also a CNN that was designed to be compact, EEGNet, https://iopscience.iop.org/article/10.1088/1741-2552/aace8c/meta, should be added to the comparisons in my view.

7. The ablation studies in table 4 are helpful, another column only removing recurrence, but keeping temporal convolution, would be informative.

8. Which projection exactly do you use for the 2d-grid EEG projection? Would be nice to include this detail.

9. Regarding the visualization, would be good to also include some caveats of what can be learned from such perturbation visualizations as mentioned inhttps://onlinelibrary.wiley.com/doi/full/10.1002/hbm.23730 and related to the issues mentioned in https://www.sciencedirect.com/science/article/pii/S1053811913010914

Minor things:
“To handle the non-differentiability of the threshold function, we used the rectangular function to approximate the gradient of a spike with respect to the membrane voltage v:” Is this a novel idea of this work or has this been done before? Would be good to cite or clarify appropriately.

 “we augmented the SNN with a temporal convolution layer and a recurrent layer.” In this part I got a bit confused during the reading, what is now part of SNN, this way it first sounded like there will be some non-SNN-layers added to the SNN. Maybe change the wording a bit to make things very clear here.

“ th_amp is the increase factor of voltage threshold” To me factor implies something is multiplied.. But I am also not a native speaker :)

---

> ### Author Response · Authors · 2022-05-31
> **Response to reviewer Gshi (1/3)**
>
> **Q.1.** In my view, to ensure that the research community can properly assess the decoding performance of spiking neural networks, there need to be direct comparisons to published results. The manuscript however only provides comparisons to reimplemented baselines “Given that the baselines employed different EEG systems requiring network architecture changes, we implemented our own versions of the baselines and optimized the hyperparameters using a separate held-out set.”... I would much favor to still see comparison to published results, on eegmmidb or also BCI Competition IV datasets, e.g. BCI Competition IV 2a… This would greatly help assess if/how big of a performance gap there is, since the compared results will have been well-tuned.
>
> **A.1.** We agree with the reviewer that comparison with published results is typically a more objective indicator of performance comparison. We clarify that we chose not to do so in this particular work, for the following reasons:
>
> * There is no single dataset that is common across all the methods. This means that, for any dataset, we would eventually need to reimplement most of the baselines, raising concerns associated with implementation issues.
> * The evaluation method that the baselines use is not leave-k-subjects-out which is what we adopted to measure the generalization ability of our method [1]. Some methods either split the entire dataset into train and test or train and evaluate on data from the same subject (Schirrmeister et al. 2017) due to which networks see trials from the same subject in training and test. Some other methods create overlapping windows because of which segments from the same trial are seen in train and test (Zhang et al. 2018). These strategies may not project directly the test regime in the real world because of which their applicability is limited.
> * We clarify that our implementation of the baselines was as well-tuned as our own method. Specifically, we performed a grid search over the hyper-parameters and the extent of the search was the same for all the methods.
>
> We really hope that, for the above reasons,the reviewer will agree with us that providing comparisons on other datasets in this particular case would not add to the impact of our results.
>
> [1] Sammut, C. & Webb, G. I. (eds) Leave-One-Out Cross-validation 600–601 (Springer, 2010).
>
> **Q.2.** Why are the comparisons on eegmmidb only binary and not multiclass?
>
> **A.2.** We clarify that the multiclass classification involves distinguishing between right hand, left hand, tongue, and both feet movements. Due to the similarities in the neural representations of the hand vs. both feet movement, and the lack of discernible signals for the tongue [1], the multiclass approach invariably decreases the performance for both our method and the baselines. Other methods compensate for this by employing a different data partitioning strategy where more information about the test trials (either same subjects or segments of same trials) is provided during training. However, with extreme training conditions- the use of just 1s of sample to train and evaluate, and leave-k-subjects-out for train-test split, the obtained accuracies for all the methods are not helpful for practical purposes. Hence, we report binary classification results which might be more useful in a practical BCI setting, in line with work, for e.g. in [2].
>
> [1] Morash, Valerie, et al. "Classifying EEG signals preceding right hand, left hand, tongue, and right foot movements and motor imageries." Clinical neurophysiology 119.11 (2008): 2570-2578.
>
> [2] Antelis, Javier M., and Luis E. Falcón. "Spiking neural networks applied to the classification of motor tasks in EEG signals." Neural networks 122 (2020): 130-143.
>
> **Q.3.** Code availability is always a plus as it allows other researchers to apply the proposed model to other datasets.
>
> **A.3.** We clarify that we have provided the code as supplementary. We have now also added a statement about the same in the revised manuscript.

---

> > ### Author Response · Authors · 2022-05-31
> > **Response to reviewer Gshi (2/3)**
> >
> > **Q.4.** Regarding the energy-efficiency/energy comparisons, I am not so sure how to interpret them due to: “ The convolutional spike encoding layer was implemented on the host machine, with the generated spikes sent to Loihi in spike-time buffers to avoid redundant communications at every timestep. The SNN decoding module was implemented on Loihi.” Is this common practice that some part of the model is not taken into account when computing energy consumption? Isn’t this comparison unfair then, assuming e.g. 2d-CNN all the energy is taken into account, while for SNN some part is left out? How big is this part, the convolutional spike encoding layer? Some clarifications here would be helpful.
> >
> > **A.4.** We thank the reviewer for their thoughtful comment. We clarify that the encoding layer adds a very small portion to the overall computational costs. If we use the number of multiplications and addition operations as a measure of computational cost, the encoding layer constitutes only 2.6% of the overall cost. We further clarify that the encoding layer can indeed be implemented on Loihi by creating encoding neurons and using EEG signals as current inputs to the neurons. The reason why we did not do so is because of the I/O bandwidth limitation of Loihi which restricts the transfer of high-precision values from the host computer to the chip. This limitation has been fixed in the second version of Loihi which will be available in the near future. Overall, not taking into consideration the encoding cost does not affect our claim of the energy efficiency of the SNNs.
> >
> > **Q.5.** Also “SNN on Loihi had reduced number of convolutional kernels (Table 1) and did not have neurons with adaptive threshold and decays.”.. Shouldn’t this then also be done for the SNN outside of Loihi?
> >
> > **A.5.** We clarify that the SNN was trained outside of Loihi with a reduced number of convolutional kernels and did not have neurons with adaptive thresholds and decays.  The trained SNN was then deployed on Loihi for energy-efficient inference.
> >
> > **Q.6.** Also for “We used layer-wise rescaling to map the full-precision weights of the trained SNN to the 8 bits integer weights on Loihi (Tang et al., 2020).”
> >
> > **A.6.** We clarify that since Loihi supports 8-bit integer weights, we could not directly deploy the full precision GPU-trained SNN on Loihi. Hence, we used a layer-wise rescaling technique for mapping the trained SNN with higher weight precision onto the chip. To do so, we rescaled the weights and voltage thresholds of each layer, while maintaining their spike outputs by fixing the weight-threshold ratio [Tang et al. 2020].
> >
> > **Q.7.** Also a CNN that was designed to be compact, EEGNet, https://iopscience.iop.org/article/10.1088/1741-2552/aace8c/meta, should be added to the comparisons in my view.
> >
> > **A.7.** We thank the reviewer for their suggestion and have added comparisons with EEGNet in the revised manuscript (Table 3, 5).
> >
> > **Q.8.** The ablation studies in table 4 are helpful, another column only removing recurrence, but keeping temporal convolution, would be informative.
> >
> > **A.8.** As per the reviewer’s suggestion, we have now added another ablation where only the recurrent layer is removed in the revised manuscript (Table 4).
> >
> > **Q.9.** Which projection exactly do you use for the 2d-grid EEG projection? Would be nice to include this detail.
> >
> > **A.9.** We clarify that the 2d projection is described in Figure 2 caption. Briefly, we grouped sensors according to their proximity to each other. Sensors that were close to each other formed the same row in the 2d matrix.
> >
> > **Q.10.** Regarding the visualization, would be good to also include some caveats of what can be learned from such perturbation visualizations as mentioned in https://onlinelibrary.wiley.com/doi/full/10.1002/hbm.23730 and related to the issues mentioned in https://www.sciencedirect.com/science/article/pii/S1053811913010914
> >
> > **A.10.** We thank the reviewer for pointing out this important detail. Per the reviewer’s suggestion, we have now added the caveat about our perturbation visualization in our revised manuscript (section 4.5). We discuss that any interpretation of the model’s behavior solely reflects the model’s behavior, and any inferences about the data itself must be carefully made.
> >
> > **Q.11.** “To handle the non-differentiability of the threshold function, we used the rectangular function to approximate the gradient of a spike with respect to the membrane voltage v:” Is this a novel idea of this work or has this been done before? Would be good to cite or clarify appropriately.
> >
> > **A.11.** We clarify that the rectangular function has indeed been proposed in prior work by [Wu et al. 2018]. Although we referenced this work in the sentence before, we have now also added the reference to it in the line under consideration, per the reviewer’s suggestion.

---

> > > ### Author Response · Authors · 2022-05-31
> > > **Response to reviewer Gshi (3/3)**
> > >
> > > **Q.12.** “we augmented the SNN with a temporal convolution layer and a recurrent layer.” In this part I got a bit confused during the reading, what is now part of SNN, this way it first sounded like there will be some non-SNN-layers added to the SNN. Maybe change the wording a bit to make things very clear here.
> > >
> > > **A.12.** We thank the reviewer for their suggestion and have accordingly modified the line in the revised manuscript to make it more clear. The revised line is: “The temporal convolution layer and the recurrent layer in the SNN captured complex non-linear temporal relationships.”
> > >
> > > **Q.13.** “ th_amp is the increase factor of voltage threshold” To me factor implies something is multiplied.. But I am also not a native speaker :)
> > >
> > > **A.13.** We thank the reviewer for the careful scrutiny of our text. Per the reviewer’s suggestion, we have now reworded the sentence in the revised manuscript for more clarity: “$th_{amp}$ is the amount by which voltage threshold increases”

---

> > > > ### Comment · Reviewer_Gshi · 2022-06-07
> > > > **Thanks for replies, some concerns remain**
> > > >
> > > > **A.1 (A.2)**
> > > > Well one can also have multiple comparisons, like one that allows comparison to at least some published numbers, and another one that is more in line with the research question. At least for supplementary would be nice to have a direct comparison to published numbers.
> > > >
> > > > **A.3.**
> > > > Thanks I was not aware. Is it planned to release the code publicly?
> > > >
> > > > **A.4.**
> > > > Thanks for clarification
> > > >
> > > > **A.5/A.6**
> > > > What I meant here was whether for the energy comparison on Table 6, the SNN (TX2) was also run with a reduced number of convolutional kernels, without neurons with adaptive thresholds and decays and with 8 bit integer weights and if not, how much of a difference this would make.
> > > >
> > > > **A.7/A.8** Thank you very much, very interesting results
> > > >
> > > > **A.9** ok so this is not based on any 2d projections used in literature like in https://arxiv.org/abs/1511.06448 ?
> > > >
> > > > **A.10-13** Thanks for clarifications

---

> > > > > ### Author Response · Authors · 2022-06-09
> > > > > **Updated response to Reviewer Gshi**
> > > > >
> > > > > **[On comparison with published results]** : Per the reviewer’s suggestion, we trained our method on the high-gamma dataset  [Schirrmeister et al. 2017] for subject 1 and obtained 87.5% accuracy vs. 89.37% for 3D-CNN. This aligns with the main results and their interpretations that we report in the paper. The observed discrepancy between our method’s accuracy and the average accuracy reported in [Schirrmeister et al. 2017] is expected, given the different experimental conditions and preprocessing steps of the data with respect to our method (see Methods Section 4.1 and 4.2). Specifically, [Schirrmeister et al. 2017] adopt a cropped training strategy, and a different preprocessing (exponential moving standardization). Other approaches entail other differences; E.g., [Zhang et al. 2018] use overlapping sliding time windows for segmentation of trials. Such differences hinder a direct comparison among methods. We did everything we could within our power and the limited time given by the review window to satisfy the reviewer’s comment. The published results use 4s long EEG trials (= 1000 timesteps when sampled at 250Hz), due to which the SNN tuning and training take a lot of time (~ a day for subject 1 on the hardware that we have, NVIDIA RTX3070 TI). Providing additional experimental evidence would shift the focus of our paper, from demonstrating the comparable performance and energy efficiency of SNNs over deep networks for EEG classification, to studying how preprocessing or data augmentation strategies can affect accuracies for specific datasets. We believe that the reviewer will agree that providing further direct comparisons with additional works that use different experimental paradigms would not add to the validity of our results. That said, we will make the code publicly available so that our methods can be freely tested for different experimental conditions and preprocessing steps.
> > > > >
> > > > > **[On the public release of code]**: Yes, we will make the code repository public after a decision on the paper has been made.
> > > > >
> > > > > **[On SNN on TX2]**: We thank the reviewer for their thoughtful remark. We clarify that the SNN on TX2 in Table 6 had reduced number of kernels, and neurons without adaptive thresholds and decays. However, it did not have 8 bit integer weights, due to which we see a slight performance difference between SNN on TX2 (79.04%) and SNN on Loihi (78.04%).
> > > > >
> > > > > **[On 2d projection]**: Yes, we clarify that our 2d projection is not based on the ones used in literature like [Bashivan et al. 2016] which uses spectral topography maps that may be computationally expensive to construct for real-time applications.

---

> > > > > > ### Comment · Reviewer_Gshi · 2022-06-13
> > > > > > **BCIC IV 2a would be a faster-to-train dataset**
> > > > > >
> > > > > > I disagree that providing direct comparisons to published results has no additional value - due to the large amount of factors influencing the results in deep learning (training hyperparameters, architecture etc.) with potentially complex interactions, results that directly compare to published results are always very valuable to me. High-gamma dataset is an interesting dataset for this, if computational time is a big problem, then BCIC IV 2a can be a faster alternative. I agree that the provided code already would help other researchers to make these comparisons.

---

> > > > > > > ### Author Response · Authors · 2022-06-13
> > > > > > > **Results on BCIC IV 2a dataset now available in supplementary**
> > > > > > >
> > > > > > > We thank the reviewer for their suggestion. While a direct fair comparison is not feasible due to different experimental conditions, we now provide a comparison of our method against the DNN methods- 3D-CNN and CNN-TC (Deep) on the BCIC IV 2a dataset in Appendix E. The results for the CNN-TC method were obtained directly from running their official code, available at https://github.com/braindecode/braindecode. We used the same code-base to run our method too and just swapped the network for training. This ensured that the data loading and preprocessing pipeline were the same for all the methods. We used the trial-wise training strategy described in [Schirrmeister et al. 2017]. Under these conditions, our method obtained an average accuracy of 48.63\%, while the CNN-TC achieved 47.87\% accuracy for the 4-class classification task.

---

### Author Response · Authors · 2022-05-31
**We thank the editor and all the reviewers for their time and thoughtful feedback**

We thank the Editor for the careful and prompt handling of our paper. We also thank all three Reviewers for their time and thoughtful feedback. We are pleased that they recognize that our paper has a “clear perspective, structure and narrative” (Reviewer 1m1u) and acknowledge the importance of our neuromorphic realization (Reviewer 1m1u, WM3g). Below, we answer specific questions and describe how we have incorporated their constructive feedback into our revised manuscript. Related pointers have been added. New text in the revised manuscript is marked in red.

---

### Decision · Action_Editors · 2022-06-14

**Recommendation:** Accept with minor revision

**Comment:**

After a careful and constructive review process with a good deal of discussion between the authors and the reviewers, 2/3 reviewers have recommended to accept. Final minor revisions to be made for the camera-ready version are:

1) All edits/editions should be converted to standard colour.
2) A link for the code should be added to the footnote on page 2 where it is currently just a place holder.
3) There was a small typo on page 6 that I detected:

"Finally, the spatiotemporal features extracted by the SNN were fed to a decoder module *comprising* of a spiking and a non-spiking fully-connected layer"

"comprising" should be "composed"

---

> ### Author Response · Authors · 2022-06-16
> **Camera-ready version has been uploaded**
>
> We thank the action editor and all the reviewers for their constructive feedback that helped improve our paper considerably. We have now uploaded the camera-ready version of the paper incorporating all the changes suggested by the editor.